# A contracting Intertropical Convergence Zone during the Early Heinrich Stadial 1

Yiping Yang [1], Lanlan Zhang [1]✉, Liang Yi[2], Fuchang Zhong[1], Zhengyao Lu [3], Sui Wan[1], Yan Du [4,5] & Rong Xiang [1]✉

Despite the fact that the response of tropical hydroclimate to North Atlantic cooling events during the Heinrich Stadial 1 (HS1) has been extensively studied in African, South American and Indonesia, the nature of such responses remains debated. Here we investigate the tropical hydroclimate pattern over the Indo-Asian-Australian monsoon region during the HS1 by integrating hydroclimatic records, and examining a $\delta^{18}O_{seawater}$ record from *Globigerinoides ruber* (white) in the tropical Indian Ocean. Our findings indicate that tropical hydrological conditions were synchronously arid in both hemispheres during the early HS1 (~18.3-16.3 ka) in the Indo-Asian-Australian monsoon region, except for a narrow, wet hydrological belt in northern low latitudes, suggesting the existence of a contracted tropical precipitation belt at that time. This study reveals that the meltwater discharge and resulting changes in global temperatures and El Niño exerted a profound influence on the tropical hydroclimate in the Indo-Asian-Australian monsoon region during the early HS1.

During the HS1 (~19–15 ka)[1], the North Atlantic region experienced a significant discharge of icebergs and a drastic reduction in the Atlantic meridional overturning circulation (AMOC). The impact of this abrupt cooling in the North Atlantic on the tropical rainfall system has been studied through the analyses of paleoclimatic records and model simulations[2,3]. Previous research has suggested that the mean position of the Intertropical Convergence Zone (ITCZ) rain-belt shifted southward in response to the cooling in the Northern Hemisphere during the HS1[4,5]. However, evidence from paleoclimatic records in southern Africa[6], the southern Indian Ocean[7–9] and the southern tropical West Pacific[10–12] has shown that severe drought conditions also existed in the southern hemisphere during the HS1 (Fig. 1). McGee et al.[13] also argued that the mean ITCZ shifts were less than 1 degree of latitude during the HS1 based on the model results. Furthermore, studies in the southern South China Sea (SCS)[14], Flores Sea[15] and Northeast Brazil[16] have revealed a two-phase structure of hydroclimatic change in the tropics during the HS1, with ITCZ rainfall strengthening (weakening) in the Early HS1 (~19.0–16.1 ka) and becoming weak (strong) during the Late HS1(~16.1–14.7 ka) in the tropical northern (southern) hemisphere.

Consequently, the direction and magnitude of the shift of the ITCZ in response to North Atlantic cooling events during the HS1 remain controversial[5]. It is increasingly challenging to explain changes in tropical hydrological climate during the HS1 solely through the mechanisms of ITCZ southward migration. Additionally, the lack of paleoclimatic records from the tropical Indian Ocean, which was influenced by ITCZ precipitation, has severely limited our understanding of the responses of tropical hydroclimate to North Atlantic cooling during the HS1.

Our reconstruction of sea surface temperature (SST) and $\delta^{18}O_{seawater}$ ($\delta^{18}O_{sw}$) changes relies on the Mg/Ca and $\delta^{18}O$ records of planktonic foraminifera *Globigerinoides ruber* sensu stricto (s.s.) obtained from a deep-sea core located in the southern Bay of the Bengal (BoB). Modern moisture flux observations show that precipitation arrives year-round at this site, with the majority occurring in the latter half of the year (May–December) (Supplementary Fig. 1), correlating with the movements of the ITCZ[17]. Hence, the location of the study site (Fig. 1, Core I106; 6°14′49.76″N, 90°00′1.04″E; 2,910 m water depth) makes it an ideal location to monitor shifts in the tropical

[1]Key Laboratory of Ocean and Marginal Sea Geology, South China Sea Institute of Oceanology, Chinese Academy of Sciences, Guangzhou 510301, China. [2]State Key Laboratory of Marine Geology, Tongji University, Shanghai 200092, China. [3]Department of Physical Geography and Ecosystem Science, Lund University, 22362 Lund, Sweden. [4]State Key Laboratory of Tropical Oceanography, South China Sea Institute of Oceanology, Chinese Academy of Sciences, Guangzhou 510301, China. [5]University of Chinese Academy of Sciences, 100049 Beijing, China. ✉e-mail: llzhang@scsio.ac.cn; rxiang@scsio.ac.cn

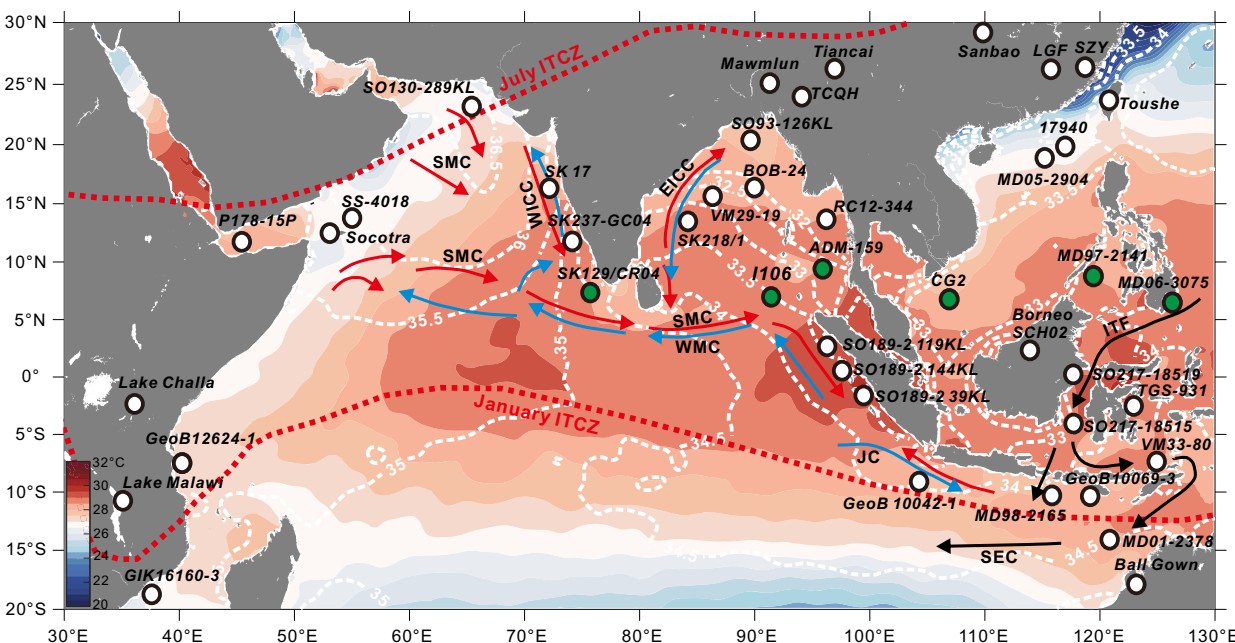

**Fig. 1 | Site map of records showing hydrological conditions during the Early Heinrich Stadial (HS1).** White dots indicate dry conditions during the Early HS1. Green dots show wet conditions during the Early HS1. Red arrows show the Summer Monsoon Current (SMC); blue arrows indicate the Winter Monsoon Current (WMC); white dashed line show the sea surface salinity (SSS). WICC West India Coastal Current, EICC East India Coastal Current, JC Java Current, ITF Indonesian Throughflow, SEC South Equatorial Current. The modern annual mean sea surface temperature (SST) and SSS distribution drawn with MATLAB software based on the World Ocean Atlas 2018 dataset[72].

rainfall belt. We assume that precipitation in our study area was mainly controlled by the Indian Ocean Summer Monsoon (IOM) and the ITCZ rain belt system during HS1[3]. We integrated the available hydroclimatic records from a latitudinal transect across the Indo-Asian-Australian (IAA) monsoon region with our results in order to evaluate the responses of the tropical hydrological cycle to the abrupt-onset HS1 cold event that occurred in the high latitudes of the Northern Hemisphere.

## Results and discussion

### $\delta^{18}O_{sw}$ reconstruction as a salinity proxy

The plankton tow samples from the study area indicate that *G. ruber* is mainly distributed in water depths of 0–50 m, and that it can therefore be classed as a mixed-layer species[18]. *G. ruber* $\delta^{18}O$ values in Core I106 become gradually negative from −1.09‰ at -24.0 ka to -2.80‰ at -1.84 ka, but exhibit an abrupt decline at 18.3–16.3 ka, with a mean value of −1.67‰ (Fig. 2). The Mg/Ca-SSTs from Core I106 show a rapid and steep increase around 19.5 ka, consistent with previous records conducted from the tropical Eastern Indian Ocean[3,19] (Fig. 2). The Mg/Ca-SST in Core I106 indicates an increase of about 0.5 °C at 16.3–18.3 ka, which corresponds to a decrease of -0.12‰ in $\delta^{18}O_{ruber}$ (assuming a change of -0.23‰ in $\delta^{18}O$ per 1 °C). Hence, the decrease in $\delta^{18}O_{ruber}$ value is primarily attributed to changes in seawater salinity in Core I106. We calculated the $\delta^{18}O_{sw}$ values of Core I106 from Mg/Ca-SST and $\delta^{18}O_{ruber}$ using the equation of Bemis et al.[20] (see "Methods"), which reflects the sea surface salinity (SSS) associated with regional hydrological changes. Similarly, the most striking characteristic of the calculated $\delta^{18}O_{sw}$ values in Core I106 is an exceptionally abrupt decline at 18.3–16.3 ka (Fig. 2).

The observed SSS and $\delta^{18}O_{sw}$ values in the southern BOB[21–23], equatorial East Indian Ocean[22], and Andaman Sea[24] demonstrate that $\delta^{18}O_{sw}$ values have a linear correlation with salinity in our study area (Supplementary Fig. 2 and Supplementary Dataset 1). Our estimates of $\delta^{18}O_{sw}$ values during the Late Holocene (2–0 ka) fall well within this linear $\delta^{18}O_{sw}$-salinity correlation (Supplementary Fig. 2).

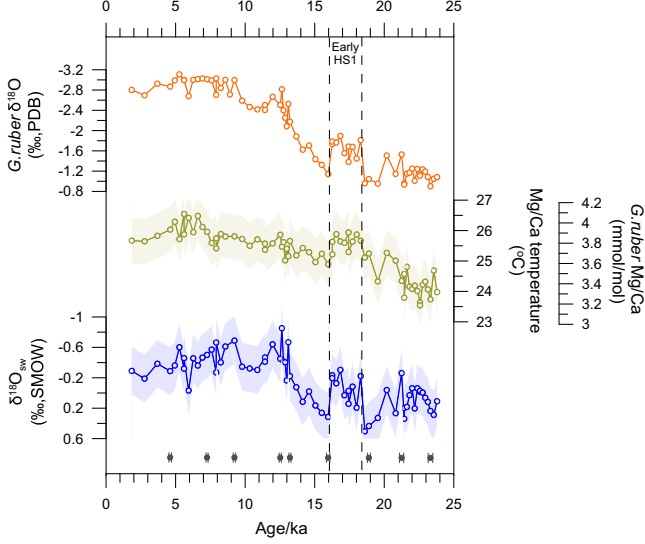

**Fig. 2 | *G. ruber* shell $\delta^{18}O$ values, Mg/Ca ratios, reconstructed Mg/Ca-temperature, and $\delta^{18}O_{seawater}$ ($\delta^{18}O_{sw}$) values from Core I106.** Shade shows one standard deviation error.

The reconstructed $\delta^{18}O_{sw}$ values for Core I106 are therefore also likely to indicate a regional SSS signal, which is related to varying quantities of fresh surface water.

### Wet hydrological conditions in the northern low latitudes during the Early HS1

Multiple $\delta^{18}O_{sw}$ records from the northern BOB[25–27] and the northern Arabian Sea[28,29], speleothem[30] and lake sediment[31,32] records from Southern China, and paleoclimatic records from the northern SCS[33], all consistently suggest that the hydrological conditions were extremely dry and long-lasting throughout the HS1 in the IAA monsoon region

(Figs. 1 and 3 and please see Supplementary Table 1). Many studies have attributed the drought conditions during the HS1 to the retraction of the Asian Summer Monsoon and the southward drift of the ITCZ, which were responses to the cooling in the North Atlantic Ocean during the HS1[34]. However, multiple paleoclimatic records from the equatorial and southern Indian Ocean[3,9,35,36] and southern Indonesia[11,12,37] also showed that dry conditions were prevalent throughout the entire HS1 period (Figs. 1 and 3). Furthermore, paleoclimatic records from Africa documented a catastrophic drought in Equatorial and Southern Africa at ~17–16 ka[6]. Therefore, the latitudinal movement of the tropical rain-belt cannot fully explain the hydroclimatic changes observed in the IAA monsoon region during the HS1.

Interestingly, our $\delta^{18}O_{sw}$ record from the tropical BoB exhibited a significant negative shift in the Early HS1 (18.3–16.3 ka), indicating a sudden decrease in SSS and an increase in fresh surface water input (Fig. 3c). The SSS in the BoB is primarily influenced by freshwater discharge and direct precipitation over the ocean[24]. However, $\delta^{18}O_{sw}$ records from the northern BOB[25–27] and lake-sediment records from southwestern China[31] revealed that there was weak monsoonal precipitation and thus reduced river runoff inflow into the BOB throughout the HS1 period, (Figs. 1 and 3b). Similarly, $\delta^{18}O_{sw}$ records from offshore Sumatra also indicated a drought during the HS1 period[3]. Therefore, the increased fresh seawater at Core I106 at about 18.3–16.3 ka was unlikely to have originated from the northern BoB or the south of Sumatra via currents. Additionally, modern hydrological data in the study area suggest that SSS is closely related to precipitation (Supplementary Fig. 1). Furthermore, the $\Delta\delta^{18}O_{ruber-dutertrei}$ archive from Core 758 (5°23.5′N,90°21.67′E), adjoining Core I106, indicated a general weakening of IOM intensity during the entire HS1[38]. This suggests that there were no significant changes in water stratification at 18.3–16.3 ka. Therefore, the changes in $\delta^{18}O_{sw}$ and SSS at Core I106 during the Early HS1 period are most likely associated with variations in tropical precipitation.

Likewise, the $\delta^{18}O_{sw}$ record from Core ADM-159 (9.27°N, 95.61°E) in the southern Andaman Sea exhibited a significant negative anomaly at about 17.0–18.7 ka[39] (Supplementary Fig. 4). The reconstructed SSS values from Core SK129/CR04 (6°29.65′N, 75°58.68′E) in the Equatorial Arabian Sea also indicated a low salinity event at 19.5–16.5 ka[40] (Fig. 3e). The $\delta^{18}O$ records of multiple planktonic foraminiferal species from the Equatorial Arabian Sea also revealed a negative peak at around 19.0–17.0 ka, which has been attributed to a stronger winter monsoon current[41,42]. However, the $\delta^{18}O_{sw}$ values in Core SK218/1 from the western BOB, which was influenced by EICC, increased significantly throughout the entire HS1, indicating a weak winter monsoon current[25] (Fig. 1). Moreover, if the winter monsoon current had strengthened, more saltwater would have been transported from the south along Sumatra into our study area; on the contrary, the $\delta^{18}O_{sw}$ values at Core I106 declined a lot during the Early HS1. We therefore suggest that the negative $\delta^{18}O$ records of planktonic foraminiferal in the Equatorial Arabian Sea during the Early HS1 may also be associated with increased tropical precipitation. Additionally, evidence provided by grain-size populations, dry bulk density, mass accumulation rates, and Si/Al ratios from Core CG2 (6.3928°N, 110.1542°E)[14] in the southern SCS suggested strong precipitation during 19.0–18.0 ka and 17.5–16.1 ka (Fig. 3d). In the Sulu Sea, the $\delta^{18}O_{sw}$ records from Core MD97-2141 (8.8°N, 121.3°E)[43] indicate that surface water in the Early HS1 was fresher than that during the Late HS1 (Fig. 3f). The X-ray fluorescence-derived log (Fe/Ca) records from MD06-3075 (6°29′N, 125°50′E) at Mindanao, which is a reliable proxy for freshwater runoff, also indicated increased precipitation at Mindanao at 15.7–17.8 ka, but with dry conditions in Borneo and China during this interval[44] (Supplementary Fig. 4). The aforementioned records from the northern low latitudes support the notion that tropical precipitation intensified significantly during the Early HS1.

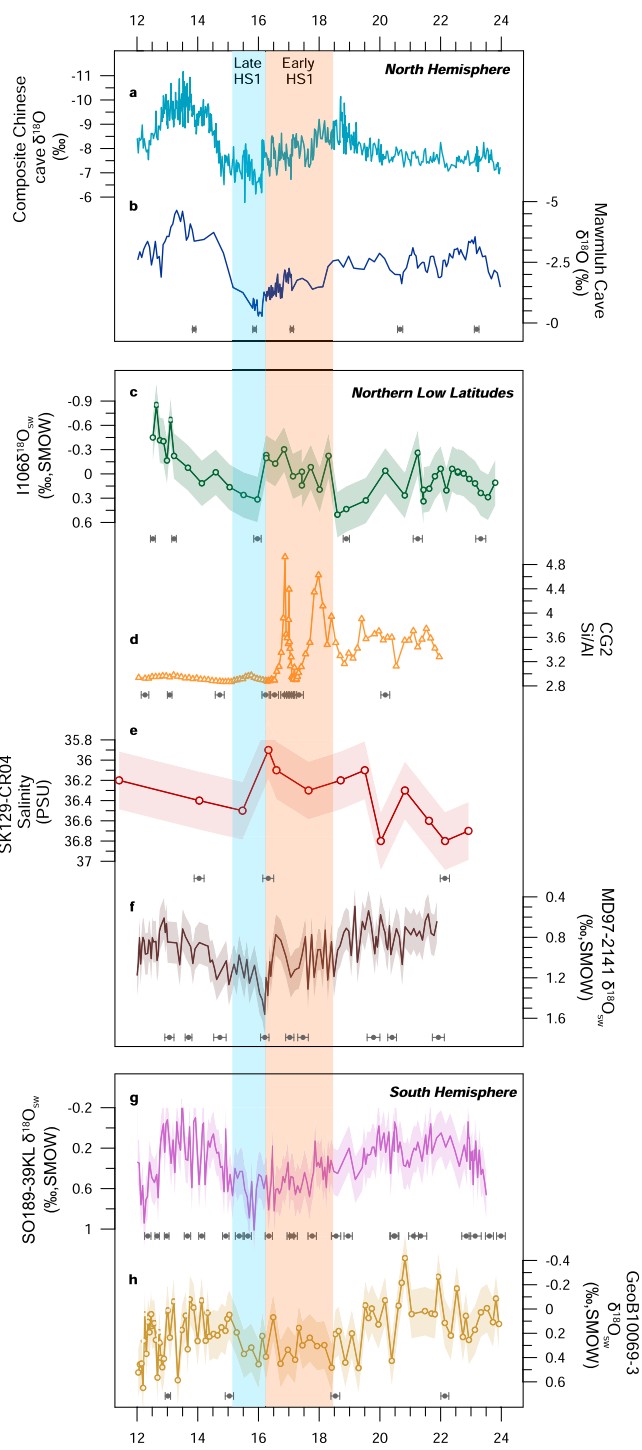

**Fig. 3 | Records showing hydrological conditions during the Early Heinrich Stadial (HS1) in the Indo-Asian-Australian (IAA) monsoon region. a** The composite Asian Monsoon $\delta^{18}O$ record[30]. **b** The Indian Ocean Summer Monsoon (IOM) proxy record from Mawmluh Cave, Meghalaya, India[27]. **c** $\delta^{18}O_{seawater}$ ($\delta^{18}O_{sw}$) records from Core I106 from the southern Bay of the Bengal (BoB) (this study). **d** Si/Al ratios from Core CG2 from the southern South China Sea (SCS)[14]. **e** Sea surface salinity (SSS) records from Core SK129-CR04 from the tropical Indian Ocean[40]. **f** $\delta^{18}O_{sw}$ records from Core MD97-2141 from the Sulu Sea[43]. **g** $\delta^{18}O_{sw}$ records from Core 189-39KL from the tropical East Indian Ocean[3]. **h** $\delta^{18}O_{sw}$ records from Core GeoB10069-3 from the Savu Sea[37]. Shade shows one standard deviation error.

Our newly-integrated paleoclimatic records from the IAA monsoon region therefore reveal that there were mostly drought hydrological conditions in both the northern and southern hemispheres in the Early HS1. However, a wet hydrological condition was identified at ~3–9°N. This evidence suggests a possible contraction of the tropical convection precipitation region during this period.

## Possible mechanisms controlling tropical hydroclimatic changes in the Early HS1

Previous studies have reported that the collapsed AMOC and cooling in the Northern Hemisphere during the HS1 resulted in an increase in interhemispheric temperature gradient, leading to a southward shift of the ITCZ[4]. Model results from the tropical East Indian Ocean suggested that there were drier conditions over the equatorial and north Indian Ocean, and more humid conditions in southern Indonesia, due to the southward displacement of the ITCZ during the HS1[3]. However, our new paleoclimatic records from the northern low latitudes support the existence of a two-phase structure of tropical hydroclimate during the HS1, with remarkable humid conditions occurring in the Early HS1. Paleoclimatic records in Core VM33-80 in south Indonesia show an arid hydrological condition in the early phase of the HS1, and a humid hydrological condition at 16–14.5 ka[15]. $\delta^{18}O_{sw}$ records in cores MD98-2165[35], MD01-2378[10,11], GeoB10069-3[37] from southern Indonesia, and stalagmite $\delta^{18}O$ record from Ball Gown[45] all indicate dry hydrological conditions in the early phase of the HS1, which is also supported by paleo-records from the southwest Indian Ocean[7,9,36] (Fig. 1). Therefore, variations in tropical precipitation patterns are not only affected by the interhemispheric temperature difference in the IAA monsoon realm, but also associated with other driving factors. In recent years, increasing evidence suggests a hemi-spherically symmetric contraction of tropical precipitation in response to glacial cycle drivers[46]. Model simulations from Africa have shown that precipitation coherency decreased in both southeastern Equatorial and Northern Africa in response to meltwater-induced reductions in the AMOC during the early phases of the last deglaciation[47]. Yan et al.[48] also pointed out that the latitudinal range of ITCZ rainfall in the Western Pacific contracted over decadal to centennial timescales in response to a cold climate during the Little Ice Age (LIA). Stalagmite record from southwest Madagascar have also shown that the tropical rain-belt simultaneously expands or contracts in both hemispheres in the past[49].

Numerous studies have reported an abrupt and early ice recession in the European Ice Sheet during the first part of HS1, leading to meltwater discharge into the Eastern North Atlantic Ocean[50,51] (Fig. 4a). This event had a significant impact on the climate both on land and in the ocean[52]. Evidence from the North Atlantic suggests that the early reduction in AMOC at ~19–16.5 ka was initiated and sustained by the enhanced melting water from Eurasian ice sheets[1]. Additionally, the melting water from the Laurentide Ice Sheet caused a further reduction in AMOC at ~16.5–15 ka[1,53,54] (Fig. 4b, c). The tropical hydroclimate within HS1, located in the northern low latitudes of the IAA monsoon region, also exhibited two distinct phases. Wet hydrological conditions were observed at about 18.3–16.3 ka, followed by dry conditions at ~16.3–14.7 ka, which was consistent with the two-step AMOC slowdown related to meltwater from different ice sheets (Fig. 4e).

During the Early HS1, the cooling of the North Hemisphere, resulting from the meltwater discharge from Eurasian ice sheets and slowdown AMOC, led to the southward migration of the westerlies[55] and restricted the northward migration of the tropical rain-belt[56]. The global surface temperature remained relatively low during this period[57,58] (Fig. 4d). At the same time, there was a sudden increase in the advection of heat toward the low latitudes of the Indian Ocean due to the anomalous transportation of heat northward into the northern high latitudes and a more vigorous ITF linked with the expansion of the

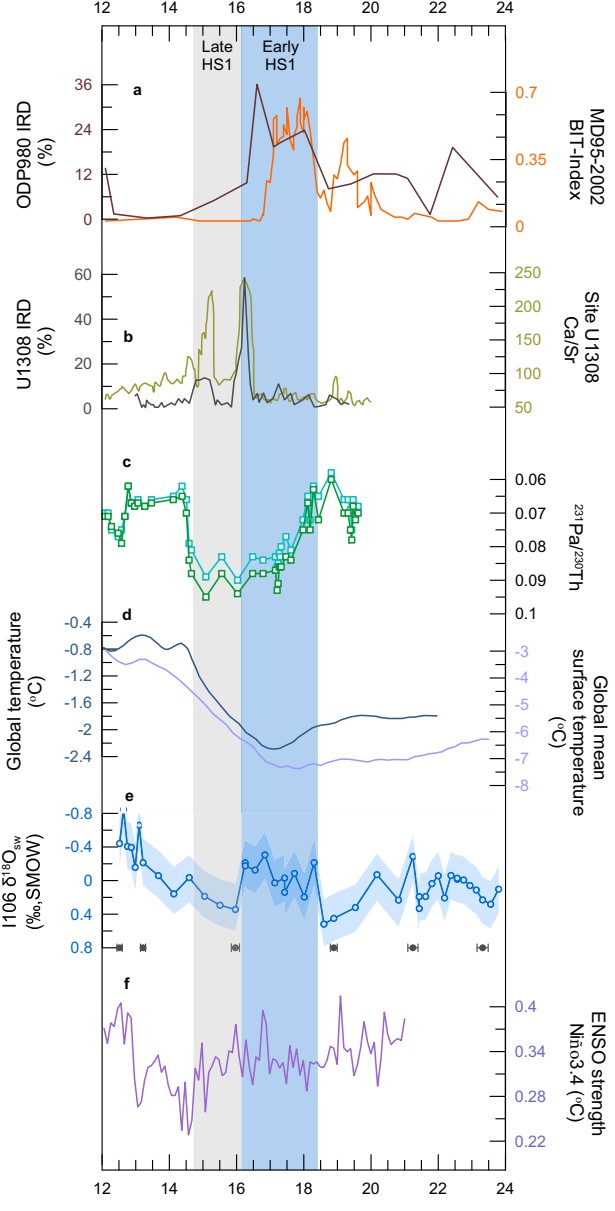

**Fig. 4 | Comparison of tropical paleoclimatic records and paleo-records of ice sheet discharge, Atlantic meridional overturning circulation (AMOC), global mean temperature, and El Niño-Southern Oscillation (ENSO) activities during the Heinrich Stadial (HS1). a** Ice-rafted debris records from Core ODP980[51], terrestrial organic matter isoprenoid tetraether (BIT) index from Core MD95-2002[50], indicating the discharge of icebergs from the Eurasian ice sheet. **b** Ice-rafted debris and Ca/Sr records from Core U1308, indicating the discharge of icebergs from the Laurentide Ice Sheet[53]. **c** $^{231}Pa/^{230}Th$ records from cores OCE326-GGC5 in the Northern Atlantic Ocean[54]. **d** Modeled global temperature stack from Shakun et al.[57] and global surface temperature from Osman et al.[58]. **e** $\delta^{18}O_{seawater}$ ($\delta^{18}O_{sw}$) records from Core I106 from the southern Bay of the Bengal (BoB) (this study). **f** ENSO variability modeled by the baseline transient simulation (TRACE)[67].

Indo-Pacific Warm Pool (IPWP)[59]. This is supported by SST records in cores I106, SO189-39KL[3], SK157-4[60], GeoB10029[19] from the low latitudes of the Indian Ocean, which suggest a steep and abrupt rise with a magnitude of >1.0 °C at about 19.5–18.0 ka, and warmer SST events around 20 ka and 17 ka from the northern Arabian Sea[61]. With enhanced tropical SST warming, the latitudinal migration of the ITCZ in the IAA monsoon region potentially reduced, especially as the seasonally-affected ITCZ generally locates over the warm ocean[62].

Collins et al.[63] have proposed that the tropical rain-belt in Africa contracted relative to the Late Holocene during the HS1, owing to a latitudinal compression of atmospheric circulation related to a lower mean global temperature. Besides, the tropical precipitation pattern in the IAA monsoon region also has a strong correlation with El Niño-Southern Oscillation (ENSO) activities[64]. Model studies indicate that there is an anticorrelation between ENSO and the Hadley circulation, which means that narrow and weak Hadley circulation occurs under El Niño condition[65]. The zonal SST difference between the West Pacific and East Pacific[66] and a transient model simulation[67] suggest a more El Niño-like state in the Early HS1 (Fig. 4f). Due to anomalous warming generated by El Niño under this state, the tropical troposphere becomes warmer, and the subtropical troposphere is cooler, which enhances the meridional temperature gradient, and then results in shrinking of the Hadley circulation in both hemispheres[68]. It was reported that ENSO variability is strongly enhanced in response to meltwater discharges and the resulting substantial slowdown of the AMOC during the Early deglaciation[67].

In summary, our research findings indicate the presence of humid conditions in the northern low latitudes, and dry hydrological conditions in both the northern and southern parts of the IAA monsoon region during the Early HS1. The synchronous occurrence of drought in both hemispheres suggests that tropical precipitation in the IAA monsoon region likely contracted latitudinally during the Early HS1. Our study demonstrates that the variability in the tropical hydroclimate pattern during the Early HS1 in the IAA monsoon region was a response to the meltwater discharge from the Eurasian ice sheet and the resulting changes in AMOC, global temperatures and El Niño. The cooling in the northern high latitudes hindered the northward expanding of the Hadley circulation, as evidenced by dry condition records in northern hemisphere during the Early HS1. Additionally, strong El Niño also led to a reduction in the extent of the Hadley circulation in the southern hemisphere[68].

## Methods

### Mg/Ca and isotope analyses

Approximately 80 *Globigerinoides ruber* sensu stricto (s.s.) individuals were selected from 250–350 μm size fractions; they were then crushed before being split into samples ready for stable isotope and Mg/Ca analysis. For Mg/Ca analysis, the pretreatment and analysis procedures followed the standard cleaning protocol developed by Barker et al.[69], including ultrasonic cleaning in alternation with washes in Milli-Q water and methanol, removal of organic matter by 2% $H_2O_2$ solution, and weak acid leaching with 0.001 M $HNO_3$. The clean samples were then dissolved in 0.075 M $HNO_3$. Samples were centrifuged to remove any remaining insoluble particles and then diluted with Milli-Q water and measured on an ICP-AES at the Key Laboratory of Ocean and Marginal Sea Geology, South China Sea Institute of Oceanology, Chinese Academy of Sciences. The instrumental precision of the ICP-AES was monitored using analysis of an external, in-house standard solution with a Mg/Ca ratio of 4.44 mmol/mol, after every three samples. The relative standard deviation of the external standard was ±0.55%. Analytical reproducibility was estimated by replicate measurements that revealed a reproducibility of Mg/Ca ±1.48% (1σ). The Mn/Ca ratio was ~0.16 mmol/mol, indicating no significant contribution of Mg from Mn-Fe-oxide coating.

For stable isotopic analysis, the shell fragments were cleaned by ultrasonication in 2% $H_2O_2$ and acetone. Stable isotopic measurements were performed on a Thermo Finnigan MAT 253 mass spectrometer with a Kiel III automatic carbonate preparation device at the Key Laboratory of Ocean and Marginal Sea Geology, South China Sea Institute of Oceanology, Chinese Academy of Sciences. The standard error of the $δ^{18}O$ was <0.05‰. Isotopic values were reported as ‰Vienna Pee Belemnite (VPDB) and calibrated with the National Bureau of Standards (NBS) 19 standards.

### Mg/Ca-SST and $δ^{18}O_{sw}$ reconstruction

Mg/Ca values were converted to temperature using the equations developed by Anand et al.[70]: Mg/Ca [mmol mol$^{-1}$] = 0.38e$^{0.09T[°C]}$. $δ^{18}O_{sw}$ values were calculated using the equation proposed by Bemis et al.[20]: T [°C] = 14.9–4.8 ($δ^{18}O_c$–$δ^{18}O_{sw}$). An additional 0.27‰ was added to them to convert the Vienna Pee Belemnite (VPDB) values to Vienna Standard Mean Ocean Water (VSMOW) values. $δ^{18}O_{sw}$ values were corrected for sea-level changes using the reconstruction protocol developed by Lambeck et al.[71].

### Error analysis for SST and $δ^{18}O_{sw}$

The errors in SST and $δ^{18}O_{sw}$ in this study was estimated using equations proposed by Mohtadi et al.[3]. The errors in SST and $δ^{18}O_{sw}$ are about ±1.03 °C and ±0.23‰, respectively. The error estimation for SST is carried out by propagating the errors introduced by the equation proposed by Anand et al.[70] and Mg/Ca measurement. The SST error estimation is given as[3]:

$$\sigma_T^2 = \left(\frac{\partial T}{\partial a}\sigma_a\right)^2 + \left(\frac{\partial T}{\partial b}\sigma_b\right)^2 + \left(\frac{\partial T}{\partial \text{Mg/Ca}}\sigma_{\text{Mg/Ca}}\right)^2 \quad (1)$$

where $a = 0.090 \pm 0.003$ °C$^{-1}$, $b = 0.38 \pm 0.02$ mmol/mol$^{-1}$, $\frac{\partial T}{\partial a} = -\frac{1}{a^2}\ln(\frac{\text{Mg/Ca}}{b})$, $\frac{\partial T}{\partial b} = -\frac{1}{ab}$ and $\frac{\partial T}{\partial \text{Mg/Ca}} = -\frac{1}{a}\frac{1}{\text{Mg/Ca}}$.

And the uncertainties in $δ^{18}O_{sw}$ is estimated by propagating errors from the $δ^{18}O$-temperature equation of Bemis et al.[20] and $δ^{18}Oc$ measurements and SST, which is given following[3]:

$$\sigma_{\delta^{18}O_{sw}}^2 = \left(\frac{\partial \delta^{18}O_{sw}}{\partial T}\sigma_T\right)^2 + \left(\frac{\partial \delta^{18}O_{sw}}{\partial a}\sigma_a\right)^2 + \left(\frac{\partial \delta^{18}O_{sw}}{\partial b}\sigma_b\right)^2 + \left(\frac{\partial \delta^{18}O_{sw}}{\partial \delta^{18}O_c}\sigma_{\delta^{18}O_c}\right)^2$$

$$(2)$$

where $a = 14.9 \pm 0.1$ °C, $b = -4.8 \pm 0.08$ °C, $\frac{\partial \delta^{18}O_{sw}}{\partial T} = -\frac{1}{b}$, $\frac{\partial \delta^{18}O_{sw}}{\partial a} = \frac{1}{b}$, $\frac{\partial \delta^{18}O_{sw}}{\partial b} = \frac{T}{b^2} - \frac{a}{b^2}$ and $\frac{\partial \delta^{18}O_{sw}}{\partial \delta^{18}O_c} = 1$.

### Chronological framework

The age model for Core I106 was determined through the utilization of mixed planktonic foraminiferal Accelerated Mass Spectrometry (AMS) radiocarbon data from 17 layers (Supplementary Table 2). Conventional $^{14}C$ ages were adjusted for isotopic fraction utilizing $δ^{13}C$ values. These ages were further calibrated into calendar ages using CALIB 8.10 software and a MARINE 20 dataset, without adjusting for a regional $^{14}C$ reservoir age. Linear interpolation was then employed to establish chronological continuity between calendar ages. The average sedimentation rate was ~6.25 cm/ka.

### Dating uncertainties

The age models utilized in this study for marine sediment records were established through the use of AMS radiocarbon dating on planktonic foraminifera. The AMS $^{14}C$ dates from marine sediment records were then converted to calendar ages using the CALIB 8.10 program and the MARINE 20 curve (please see Supplementary Dataset 2). The age models for terrestrial records referenced in this study were revised using the IntCal 20 curve instead of the Marine 20 curve. These age models were created through linear interpolating between derived intermediate calendar ages. It is important to note that a regional $^{14}C$ reservoir age was not applied to all cores in this study. The revised dates are listed in the Supplementary Material. The stalagmite $δ^{18}O$ records were dated using $^{234}U/^{230}Th$ measurements, as described in the original paper.

### Collection of existing paleoclimatic records

Numerous paleoclimate records of the HS1 have been documented in the IAA monsoon regions. In this study, we gathered 43 records that possess four AMS$^{14}C$ age control points ranging from 12 to 24 ka (Supplementary Table 1). The temporal resolution of each sample is

generally superior to 500 years, except SK129/CR04, which has three AMS $^{14}$C with 700 years per sample. Additionally, we collected seven δ$^{18}$O$_{sw}$ records from the northeast Indian Ocean, seven paleoclimatic records from the northern Arabian Sea, two paleoclimate records from the northern SCS, six paleoclimate records from the southern part of China and the Indian subcontinent, one paleoclimate record from Taiwan, one record from the southern SCS, one record from the Sulu Sea, one record from Mindanao, four records from the southern Arabian Sea and Southern Africa, eleven paleoclimate records and two stalagmite δ$^{18}$O records from the south tropical Indian Ocean and tropical West Pacific. These records were collected to represent the overall spatial distribution pattern of hydrological conditions during the HS1 ranging from 30° north to 20° south in the IAA Monsoon region.

## Data availability

Data generated in this study are available in Pangaea repository https://doi.org/10.1594/PANGAEA.956013. Source data are provided with this paper.

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

## Acknowledgements

This research received financial support from the National Natural Science Foundation of China (Grant No. 42176082 to R.X., 41906057 to Y.Y., 42176080 to L.Z., 41876056 to L.Z., 41576044 to L.Z.), the Natural Science Foundation of Guangdong Province (Grant No. 2021A1515011501 to Y.Y., 2023A1515010640 to S.W.), the development fund of South China Sea Institute of Oceanology of the Chinese Academy of Sciences (Grant No. SCSIO202201 to Y.D. and L.Z.), and the basic research plan and applied basic research project of Guangzhou (Grant No. 202201010624) to Y.Y., Additional funding was provided by the Swedish Research Council Vetenskapsrådet (Grant No. 2022-03617) to Z.L. We thank the captain, crew and scientists onboard the R/V ShiYan 3 for their efforts in collecting samples during cruises of the Eastern Indian Ocean Comprehensive Scientific Expedition in Spring 2017 (No. 41649910).

## Author contributions

Y.Y. proposed this idea and contributed experimental analysis, writing—original draft and revising manuscript. L.Z. contributed to sample, funding acquisition and revising manuscript. L.Y. contributed to funding acquisition and validation. F.Z. contributed to experimental analysis, data analysis. Z.L. contributed to data analysis and revising manuscript.

S.W. contributed to data curation and revising manuscript. Y.D. contributed to hydrological data analysis, mapping and revising manuscript. R.X. contributed to writing—review and editing, supervision and validation and revising manuscript. All authors contributed to analyzing and discussing the results, and editing and revising the manuscript.

## Competing interests

The authors declare no competing interests.
