## [Peer Review File · Nature Communications]

A contracting Intertropical Convergence Zone during the Early Heinrich Stadial 1Reviewer #1 (Remarks to the Author):

Review: A contracting Intertropical Convergence Zone during the Early HS1

This manuscript presents new data and synthesizes a large amount of existing data to make the compelling case that the ITCZ contracted to a narrow zonal band across much of the Indian and western-most Pacific, consistent with previously published hypotheses regarding ITCZ dynamics over Africa and western Pacific at different time intervals. The new record (1106) is well dated and has compelling structure consistent with the authors interpretation thereof. The cores to which it is compared also have well-defined structure that the authors synthesize and interpret in a consistent, well-reasoned, and clearly presented manuscript.

As described in detailed comments below, the manuscript would benefit from a Sup Mat section that provides age-model details of the existing cores used in the transect, specifically indicating the age control governing the very short interval of interest in this work (early HS1; 18.7 to 16.5 ka). Is the age control in the cores synthesized sufficient to document the synchronous nature of the event seen in 1106 and interpreted as a contracted ITCZ at the narrow latitudinal band inferred?

Specific comments

14 Despite the responses of the tropical hydroclimate to a North Atlantic cooling event during the Heinrich

15 Stadial 1 (HS1) having been extensively studied in African, South American and Indonesian, the nature of

16 such responses remains many debates.

Intent is clear – edit for grammar.

This

24 study reveals that variable, millennial-scale seasonal insolation patterns, and low global temperatures

25 associated with meltwater discharge from the Eurasian ice sheet, both exerted a profound influence on the

26 tropical hydroclimate during the HS1.

Discharge from the Eurasian ice sheet into what region? Please specify where the discharge is going- eastern N. Atl?

It is not clear what 'variable millennial-scale seasonal insolation patterns' means. Please clarify. Secular or orbital variations?

49 Our reconstruction of SST and residual $\delta^{18}\text{O}_{\text{seawater}}$ ($\delta^{18}\text{O}_{\text{sw-residual}}$)

Residual relative to what? 'Residual' implies something has been subtracted from the $\delta^{18}\text{O}_{\text{sw}}$ record. If not, then consider dropping the modifier 'residual'. Clarify how $\delta^{18}\text{O}_{\text{sw}}$ is constructed.

Modern moisture flux observations show that

52 precipitation arrives year-round at this site, with the majority sourced from the local tropical Eastern Indian

53 Ocean during the summer months, correlating with the movements of the ITCZ **12**. Hence, the location of the

54 study site (Fig. 1, Core I106; 6°14'49.76"N, 90°00'1.04"E; 2,910 m water depth)

**makes it an ideal location
55 to monitor shifts in the tropical rainfall belt.**

Please justify assumption that modern moisture source areas and transport paths at the core location also apply during HS1, making the site appropriate for use in this study.

56 latitudinal transect across the Asian-Indian Ocean

An odd geographic term. There is an Indian Ocean but not an Asian Ocean. How about the Indo-Asian-Australian (IAA) monsoon region? It's a difficult area to define geographically for sure.

the calculated 18Osw

73 and 18Osw-residual values is an exceptionally abrupt decline at 18.7-16.5 ka BP (Fig. 2).

The reader has not been introduced to what these two terms mean and does not yet know the difference between them, leading to confusion. Please define them where each is first used or point the reader to the methods section or sup mat section where they are clearly defined.

401 Chronological framework. The age model for Core I106 was established using mixed planktonic foraminiferal AMS

402 radiocarbon data from 17 layers (Table S2). Conventional 14C ages were corrected for isotopic fraction using $\delta^{13}C$ values.

403 Conventional 14C ages were calibrated into calendar ages using BetaCal software and a MARINE13 dataset, adjusting for a

404 regional 14C reservoir age (global reservoir age = 400 yr, $\Delta R = 32 \pm 70$ yr). The mean sedimentation was ~ 6.25 cm/ka.

This age model appears robust. Are the age models from the other cores used in comparison similarly robust? See next comment.

The results show that tropical

21 hydrological conditions were synchronously arid in both hemispheres during the early HS1 (18.7-16.5 ka

22 BP) in the Asian-Indian Monsoon region, except for a narrow, wet hydrological belt in northern low latitudes,

23 suggesting the existence of a contracted Intertropical Convergence Zone (ITCZ) rain-belt at that time.

18.7 to 16.5 ka is a very short interval of time. This paper does not (yet) present sufficient information to demonstrate that this short interval is sufficiently resolve in all the cores utilized. In order to draw the spatial-temporal conclusions in this paper, the age models at all the sites used need to be evaluated to demonstrate that they identify/resolve the interval of interest sufficiently.

For example, 1106 (this study) has six dates in the interval depicted in figure 2. It is well resolved. Are the other six cores sufficiently dated as well; sufficient to draw the conclusions presented in the manuscript? This important topic could be addressed in the Sup Mat.

86 Figure 2. G. ruber shell $\delta^{18}O$ values, Mg/Ca ratios, reconstructed Mg/Ca-temperature

ratios, $\delta^{18}\text{O}$ seawater values, and residual
87 $\delta^{18}\text{O}$ seawater values from Core I106

Propagated error in the x and y dimensions would be useful in figure 2

This would indicate that tropical precipitation fell
within a narrower geographical 132 range during the Early HS1 than during the
Holocene.

'...HS1 compared to the Holocene'?

135 Figure 3. Records showing wet hydrological conditions during the Early HS1 in the
tropical Asian-Indian Monsoon region.

136 (A) *G. ruber* $\delta^{18}\text{O}$ records from the Equatorial Indian Ocean 40. (B) Si/Al ratios
from Core CG2 from the southern SCS 6. (C)

137 $\delta^{18}\text{O}_{\text{sw}}$ -residual records from Core I106 from the southern BoB (this study). (D)
 $\delta^{18}\text{O}_{\text{sw}}$ -residual records from Core SK129-CR04

138 from the tropical Indian Ocean 42. (E) $\delta^{18}\text{O}_{\text{sw}}$ -residual records from Core MD01-
2390 from the southern SCS 43.

The location of AMS dates on the various cores would be useful on Figures 2, 3, and 4.
This would help the reader assess the extent to which the structures in the various cores
align or not.

During the early phases of the last deglaciation, boreal insolation was lowest during the
summer months, and the ITCZ rain-belt was therefore drawn toward 147 the Equator;
however, boreal winter

148 insolation reached its maximum during this period, and the ITCZ rain-belt migrated
further north (Fig. 4D). 149 Consequently, the seasonal migration range of the tropical
rain-belt during the Early HS1 was probably

150 relatively narrow

DJF and JJA insolation should probably be plotted on the same Y-axis scale. The
differing scales gives the impression that DJF and JJA are similarly strong when, in
comparison, JJA has a larger absolute value and larger range. It is not clear that this
forcing would substantially narrow the ITCZ rain belt.

176 In conclusion, our findings highlight the presence of dry hydrological conditions in
both the northern

177 and southern parts of the Asian-Indian Monsoon region during the Early HS1. These
dry conditions cannot

178 be explained by a simple southward drift of the tropical rain-belt. Analyzing paleo-
records from the northern

179 low latitudes in the Asian-Indian Monsoon region, we established that a narrow
tropical rain-belt existed

180 during the Early HS1. This study presents a clear picture of the variations in
hydrological conditions during

181 the Early HS1 in the Asian-Indian Monsoon region.

The last sentence might be replaced by a summary of the mechanisms driving the
narrowed ITZC from both Northern- and Southern-Hemisphere perspectives. This would

make for a considerably stronger conclusion statement and enhance the relevance to the wider scientific community targeted by Nature Communications.

Reviewer #2 (Remarks to the Author):

Yang and others focus on the response of the tropical Intertropical Convergence Zone (ITCZ) feature over the Indian Ocean during Heinrich Stadial 1 (HS1) by presenting a new record of seawater- $\delta^{18}\text{O}$ (a proxy for salinity; $\delta^{18}\text{O}_{\text{sw}}$) from foraminiferal geochemistry in marine sediments proximal to the Ninetyeast Ridge and by compiling a series of previously published records. Their record shows lower $\delta^{18}\text{O}_{\text{sw}}$ (or lower surface-salinity) during the initial part of HS1 and Yang and colleagues suggest that this is in accordance with a series of other records showing intensified rainfall offshore the tip of peninsular India & in the northern western Pacific Warm Pool region. These records are in contrast to widespread dual-hemispheric and pan-Indian Ocean records of drought conditions during HS1. Thus, the authors argue that the ITCZ was restricted to a narrow band during the early part of HS1 and that rainfall in this contracted ITCZ was more intense. Whereas the premise is interesting and their record will certainly be of value to the paleoceanographic community, I am unable to recommend this manuscript for publication in Nature Communications as I am yet unconvinced by their mechanistic arguments. I detail my points below:

Proxy uncertainties and evidence for a "wetter" yet narrower ITCZ band during the early part of HS1: I am not convinced of the purported "wetter" records that are presented in the manuscript (Fig. 3 & Table S1). For one, it is extremely challenging to reconstruct precipitation using surface-ocean $\delta^{18}\text{O}$ and at the very least, the authors must propagate the uncertainties appropriately from paired Mg/Ca- $\delta^{18}\text{O}$ data (see e.g. Rohling, 2000; Thirumalai et al. 2016; Gray & Evans, 2019 etc.). Thus, it is unclear whether the changes shown in Fig. 3 are actually consistent with less saline conditions during HS1, given that anomalies are on the order of $\sim 0.1\text{--}0.3\text{‰}$ across all records. Please perform a robust error propagation exercise to see whether such an anomaly persists. Moreover, the authors suggest that they only use the Dekens et al. 2002 equation for inverting SSTs - they must additionally test for the influence of salinity (e.g. see Gray & Evans, 2019) on Mg/Ca and see whether their result holds. They should perform the same analysis across all purported records consistent with "fresher" conditions. Finally, the Sulu Sea & Mindanao Dome records seem to show a steady increase in $\delta^{18}\text{O}_{\text{sw}}$ over the early part of HS1 and contrast with the record from Core I106; yet this discrepancy is not explained. Moreover, I do not know if the average $\delta^{18}\text{O}_{\text{sw}}$ from the LGM to the early part of HS1 in these records (Rosenthal et al. 2003; Boillet et al. 2011) are even significantly different from one another. This needs to be tested within uncertainty.

Winter Monsoon Rainfall: Thus, it appears that only the I106 record seems to show a structured anomalous response of lower $\delta^{18}\text{O}_{\text{sw}}$ in the sub-selected 7 records purported to show higher rainfall during the early part of HS1 (although the robustness is to be tested as indicated above). Whereas the Sulu Sea & Mindanao records response are uncertain, in the Indian Ocean side, the CR04 (Fig. 3D) also seems to be uncertain and not robust in its response. This leaves potentially the SS3827G record - which is only a $\delta^{18}\text{O}$ -calcite record without a temperature measurement. Thus it is also uncertain in its hydroclimatic response. Nevertheless, if this lower $\delta^{18}\text{O}$ -calcite was indeed caused by lower $\delta^{18}\text{O}_{\text{sw}}$, then the authors need to address the idea that this is due to the advection of fresh runoff from the southeastern Indian coast due to strengthened winter monsoon rainfall (see e.g. Sarkar et al. 1990 or Kumar & Ramesh, 2017). Is it possible that these low salinity waters also made their way towards the core site? That said, the authors do not provide enough clarity to rule out that these different core sites might be representing rainfall/salinity during different parts of the year. This dimension needs to

be discussed in a revised manuscript.

Forcing Mechanisms: The proposed mechanisms for the interaction of orbital forcing with abrupt climate change do not take into account the latest results suggesting that orbital forcing modulates millennial-scale activity. That said, I was confused while reading the mechanism because the authors do not explain what happens to the ITCZ in the southern hemisphere of the tropical Indian Ocean, which when DJF insolation is relatively higher, should exhibit a seasonally more southern ITCZ in the Southern Hemisphere. How does this affect their proxy comparison?

Comparisons with models: The authors do not present comparisons with climate model output - which by itself is not a problem. However, there are several papers (e.g. Mohtadi et al. 2014) which use model simulations and show that there is evidence for ITCZ movement but not intensification - let alone intensification associated with a contraction. Can the authors speculate as to why this may be the case? Can they rule out that the seasonal-bias of different proxies affects this finding and that it can be applied to mean-annual ITCZ shifts?

References:

Gray, W. R., & Evans, D. (2019). Nonthermal Influences on Mg/Ca in Planktonic Foraminifera: A Review of Culture Studies and Application to the Last Glacial Maximum. *Paleoceanography and Paleoclimatology*, 34(3), 306–315. <https://doi.org/10.1029/2018pa003517>

Kumar, P. K., & Ramesh, R. (2017). Revisiting reconstructed Indian monsoon rainfall variations during the last ~25ka from planktonic foraminiferal $\delta^{18}\text{O}$ from the Eastern Arabian Sea. *Quaternary International*, 443, 29–38. <https://doi.org/10.1016/j.quaint.2016.07.012>

Mohtadi, M., Prange, M., Oppo, D. W., De Pol-Holz, R., Merkel, U., Zhang, X., Steinke, S., & Lückge, A. (2014). North Atlantic forcing of tropical Indian Ocean climate. *Nature*, 509(7498), 76–80. <https://doi.org/10.1038/nature13196>

Rohling, E. J. (2000). Paleosalinity: confidence limits and future applications. *Marine Geology*, 163(1–4), 1–11. [https://doi.org/10.1016/s0025-3227\(99\)00097-3](https://doi.org/10.1016/s0025-3227(99)00097-3)

Rosenthal, Y., Oppo, D. W., & Linsley, B. K. (2003). The amplitude and phasing of climate change during the last deglaciation in the Sulu Sea, western equatorial Pacific. *Geophysical Research Letters*, 30(8), 1428–1424. <https://doi.org/10.1029/2002GL016612>

Sarkar, A., Ramesh, R., Bhattacharya, S. K., & Rajagopalan, G. (1990). Oxygen isotope evidence for a stronger winter monsoon current during the last glaciation. *Nature*, 343(6258), 549–551. <https://doi.org/10.1038/343549a0>

Thirumalai, K., Quinn, T. M., & Marino, G. (2016). Constraining past seawater $\delta^{18}\text{O}$ and temperature records developed from foraminiferal geochemistry. *Paleoceanography*. <https://doi.org/10.1002/2016PA002970>

REVIEWER COMMENTS

Reviewer #1 (Remarks to the Author):

Review: A contracting Intertropical Convergence Zone during the Early HS1

This manuscript presents new data and synthesizes a large amount of existing data to make the compelling case that the ITCZ contracted to a narrow zonal band across much of the Indian and western-most Pacific, consistent with previously published hypotheses regarding ITCZ dynamics over Africa and western Pacific at different time intervals. The new record (1106) is well dated and has compelling structure consistent with the authors interpretation thereof. The cores to which it is compared also have well-defined structure that the authors synthesize and interpret in a consistent, well-reasoned, and clearly presented manuscript. As described in detailed comments below, the manuscript would benefit from a Sup Mat section that provides age-model details of the existing cores used in the transect, specifically indicating the age control governing the very short interval of interest in this work (early HS1; 18.7 to 16.5 ka). Is the age control in the cores synthesized sufficient to document the synchronous nature of the event seen in 1106 and interpreted as a contracted ITCZ at the narrow latitudinal band inferred?

Thank you very much for your constructive comments. We have revised the manuscript accordingly. Our point-by-point responses are detailed below.

The questions about the age-model details of the existing cores above have been answered in detail in **Comment #8** below.

Specific comments

1. Despite the responses of the tropical hydroclimate to a North Atlantic cooling event during the Heinrich Stadial 1 (HS1) having been extensively studied in African, South American and

Indonesian, the nature of such responses remains many debates. Intent is clear -edit for grammar.

[Response]

Thank you very much for pointing out this error.

We have modified this sentence's grammar, please see **Line 14-16**.

2. This study reveals that variable, millennial-scale seasonal insolation patterns, and low global temperatures associated with meltwater discharge from the Eurasian ice sheet, both exerted a profound influence on the tropical hydroclimate during the HS1.

Discharge from the Eurasian ice sheet into what region? Please specify where the discharge is going - eastern N. Atl?

[Response]

Thanks for your suggestion. Yes, the region is Eastern North Atlantic. We have specified that an abrupt and early Eurasian fluvial discharge into the Eastern North Atlantic in **Line 161-163**.

3. It is not clear what 'variable millennial-scale seasonal insolation patterns' means. Please clarify. Secular or orbital variations?

[Response]

Thank you very much for your comment.

This term might be not accurate and distractive. The low global temperature, warming SST in tropical Indian Ocean and ENSO activities play a more important role in the contraction of tropical precipitation in Asian Monsoon region than the seasonal insolation pattern during the Early HS1 (**Line 191-216**), thus **we removed this term in the revised manuscript**.

4. Our reconstruction of SST and residual $\delta^{18}\text{O}_{\text{seawater}}$ ($\delta^{18}\text{O}_{\text{sw-residual}}$). Residual relative to what? 'Residual' implies something has been subtracted from the $\delta^{18}\text{O}_{\text{sw}}$ record. If not, then consider dropping the modifier 'residual'. Clarify how $\delta^{18}\text{O}_{\text{sw}}$ is constructed.

[Response]

Thanks for your suggestion.

The stable oxygen isotopic composition ($\delta^{18}\text{O}$) of planktonic foraminiferal shells is mainly influenced by seawater temperature and Seawater $\delta^{18}\text{O}$ ($\delta^{18}\text{O}_{\text{sw}}$). We calculated the $\delta^{18}\text{O}_{\text{sw}}$ value based on the $\delta^{18}\text{O}$ of planktonic foraminiferal and Mg/Ca-SST value from the same sample using the equation of Bemis (1998). $\delta^{18}\text{O}_{\text{sw}}$ signal reflects both salinity of the water mass associated with local factors (i.e. runoff, precipitation) and global ice volume. In order to get the salinity of the water mass associated with local factors, the effect of global ice volume should be subtracted from $\delta^{18}\text{O}_{\text{sw}}$, indicating the local seawater salinity signal. **In this revised manuscript, in order not to confuse the reader, we have corrected $\delta^{18}\text{O}_{\text{sw-residual}}$ to $\delta^{18}\text{O}_{\text{sw}}$, and clarified how $\delta^{18}\text{O}_{\text{sw}}$ was constructed in Methods part (Line 257-261).**

5. Modern moisture flux observations show that precipitation arrives year-round at this site, with the majority sourced from the local tropical Eastern Indian Ocean during the summer months, correlating with the movements of the ITCZ. Hence, the location of the study site (Fig. 1, Core I106; 6°14'49.76"N, 90°00'1.04"E; 2,910 m water depth) makes it an ideal location to monitor shifts in the tropical rainfall belt.

Please justify assumption that modern moisture source areas and transport paths at the core location also apply during HS1, making the site appropriate for use in this study.

[Response]

Thanks for your suggestion.

Modern moisture flux observations show that precipitation arrives year-round at this site, and the majority occurs in the summer months (Supplementary Fig. 1), correlating with the movements of the ITCZ (Belgaman et al., 2017) (**Line 51-53**). Hence, the location of the study site makes it an ideal location to monitor shifts in the tropical rainfall belt.

We make the assumption that the hydrologic cycle in this region is largely consistent during HS1 and present day based on the following situation: 1) marine and continental tectonic during the HS1 is consistent with it today, and the structure of land-sea thermodynamic between the Indian Ocean and Asian Continent has not changed significantly; 2) there also existed Indian Ocean Summer

Monsoon and ITCZ rain belt system during the HS1. Accordingly, we assume that precipitation in our study area in the HS1 was still linked with Indian Ocean Summer Monsoon and the ITCZ rain-belt system; 3) **in addition, other paleoclimatic studies also suggest that tropical East Indian Ocean was affected by the Indian Ocean Summer Monsoon and ITCZ rain belt system in the HS1** (Mohtadi et al., 2010, 2011, 2014). These justifications have been added to the main text **(Line 53-57)** .

6. latitudinal transect across the Asian-Indian Ocean, An odd geographic term. There is an Indian Ocean but not an Asian Ocean. How about the **Indo-Asian-Australian (IAA) monsoon region**? It's a difficult area to define geographically for sure.

[Response]

Thanks for your suggestion.

Indo-Asian-Australian monsoon region is a good definition. We have changed “Asian-Indian Ocean” into “Indo- Asian-Australian monsoon region” in the revised manuscript **(Line 57-60)**.

7. the calculated $\delta^{18}\text{O}_{\text{sw}}$ and $\delta^{18}\text{O}_{\text{sw-residual}}$ values is an exceptionally abrupt decline at 18.7-16.5 ka BP (Fig. 2). The reader has not been introduced to what these two terms mean and does not yet know the difference between them, leading to confusion. Please define them where each is first used or point the reader to the methods section or sup mat section where they are clearly defined.

[Response]

Thank you for your suggestion.

Following your **Comment #4**, in this revised manuscript, we have corrected $\delta^{18}\text{O}_{\text{sw-residual}}$ to $\delta^{18}\text{O}_{\text{sw}}$, and clarified how $\delta^{18}\text{O}_{\text{sw}}$ was constructed in Methods part **(Line 256-262)**.

8. Chronological framework. The age model for Core I106 was established using mixed planktonic foraminiferal AMS radiocarbon data from 17 layers (Table S2). Conventional ^{14}C ages were corrected for isotopic fraction using $\delta^{13}\text{C}$ values. Conventional ^{14}C ages were calibrated into calendar ages using BetaCal software and a MARINE13 dataset, adjusting for a regional ^{14}C

reservoir age (global reservoir age = 400 yr, $\Delta R = 32 \pm 70$ yr). The mean sedimentation was ~ 6.25 cm/ka. This age model appears robust. Are the age models from the other cores used in comparison similarly robust? See next comment.

The results show that tropical hydrological conditions were synchronously arid in both hemispheres during the early HS1 (18.7-16.5 ka BP) in the Asian-Indian Monsoon region, except for a narrow, wet hydrological belt in northern low latitudes, suggesting the existence of a contracted Intertropical Convergence Zone (ITCZ) rain-belt at that time. 18.7 to 16.5 ka is a very short interval of time. This paper does not (yet) present sufficient information to demonstrate that this short interval is sufficiently resolved in all the cores utilized. In order to draw the spatial-temporal conclusions in this paper, the age models at all the sites used need to be evaluated to demonstrate that they identify/resolve the interval of interest sufficiently.

For example, 1106 (this study) has six dates in the interval depicted in figure 2. It is well resolved. Are the other six cores sufficiently dated as well; sufficient to draw the conclusions presented in the manuscript? This important topic could be addressed in the Sup Mat.

[Response]

We are extremely grateful to the reviewer for pointing out this issue.

Firstly, we have re-sifted the all cores utilized in this study. We set a requirement for at least four age control points between 12 and 24 ka, and the temporal resolution must be greater than 500 yr/sample, except for Core SK129/CR04 with 3 age control points and ~ 700 yr/sample (**Line 272-293**).

Secondly, we have revised the age models at all the sites used in this study as you suggested. For the marine sediment records, we reconstructed the age models by the CALIB 8.10 program and the Marine 20 curve which are all based on Accelerated Mass Spectrometry (AMS) radiocarbon dates measured on planktonic foraminifera. For terrestrial records, the age models were revised with the IntCal 20 curve instead of the Marine 20 curve. Note that we did not apply a regional ^{14}C reservoir age in all the cores in this study. All dates are listed in supplementary material. And the stalagmite $\delta^{18}\text{O}$ records are dated using $^{234}\text{U}/^{230}\text{Th}$ measurements as in their original paper (**Line 272-293**) (**Please see supplementary material for age models**).

Age information for wet condition records in north of Equator

Core	Wet condition event	Age control points (12-24 ka)	temporal resolution	proxy
I106	16.3-18.3 ka	6	270 yrs/sample	$\delta^{18}\text{O}_{\text{sw}}$
CG2	16.6-17.2 ka, 17.4-18.5 ka	11	95 yrs/sample	Si/Al/grain size
ADM-159	16.5-18.7 ka	4	240 yrs/sample	$\delta^{18}\text{O}_{\text{sw}}$
MD97-2141	16.4-18.7 ka	9	73 yrs/sample	$\delta^{18}\text{O}_{\text{sw}}$
MD06-3075	15.7-17.9 ka	4	60 yrs/sample	Log(Fe/Ca)
SK129/CR04	16.3-19.5 ka	3	700 yrs/sample	$\delta^{18}\text{O}_{\text{sw}}$

Thirdly, in order to make our results more reliable, we also test the **validity of the reconstructed Mg/Ca-SST and $\delta^{18}\text{O}_{\text{sw}}$ of cores I106, ADM-159, MD97-2141, SK129/CR04 using the PSU solver with two sigma analytical uncertainty of Mg/Ca and $\delta^{18}\text{O}$ and $\pm 5\%$ age uncertainty**. And the results show that all $\delta^{18}\text{O}_{\text{sw}}$ demonstrate robust negative signals in the early Heinrich stadial 1, as well as original dates (**Supplementary Line 56-81**).

9. Figure 2. *G. ruber* shell $\delta^{18}\text{O}$ values, Mg/Ca ratios, reconstructed Mg/Ca-temperature ratios, $\delta^{18}\text{O}_{\text{seawater}}$ values, and residual $\delta^{18}\text{O}_{\text{seawater}}$ values from Core I106 Propagated error in the x and y dimensions would be useful in figure 2.

[Response]

Thanks for your suggestion.

We have add the propagated errors in the X and Y demensions in Figure 2 and Figure 3.

10. This would indicate that tropical precipitation fell within a narrower geographical range during the Early HS1 than during the Holocene. ‘...HS1 compared to the Holocene’?

[Response]

Thank you for your comment.

We have modified this sentence, please see **Line 145-148**.

11. Figure 3. Records showing wet hydrological conditions during the Early HS1 in the tropical Asian-Indian Monsoon region. (A) *G. ruber* $\delta^{18}\text{O}$ records from the Equatorial Indian Ocean 40. (B) Si/Al ratios from Core CG2 from the southern SCS 6. (C) $\delta^{18}\text{O}_{\text{sw-residual}}$ records from Core I106 from the southern BoB (this study). (D) $\delta^{18}\text{O}_{\text{sw-residual}}$ records from Core SK129-CR04 from the tropical Indian Ocean 42. (E) $\delta^{18}\text{O}_{\text{sw-residual}}$ records from Core MD01-2390 from the southern SCS.

The location of AMS dates on the various cores would be useful on Figures 2, 3, and 4. This would help the reader assess the extent to which the structures in the various cores align or not.

[Response]

We are grateful for your suggestion.

We have added the AMS dates on the various cores in Figure 2, 3 and 4.

12. During the early phases of the last deglaciation, boreal insolation was lowest during the summer months, and the ITCZ rain-belt was therefore drawn toward the Equator; however, boreal winter insolation reached its maximum during this period, and the ITCZ rain-belt migrated further north (Fig. 4D). Consequently, the seasonal migration range of the tropical rain-belt during the Early HS1 was probably relatively narrow. DJF and JJA insolation should probably be plotted on the same Y-axis scale. The differing scales gives the impression that DJF and JJA are similarly strong when, in comparison, JJA has a larger absolute value and larger range. It is not clear that this forcing would substantially narrow the ITCZ rain belt.

[Response]

Thanks for your comment.

Based on climatic model results, Singarayer et al. (2017) have proposed that there was ocean dominated expansion and contraction of the tropical rainbelt during the late Quaternary. This expansion/contraction is the result of the different response of the marine ITCZ when at its northern and southern extremes. In boreal summer, when ITCZ is farthest north, if the insolation is the lowest, the ITCZ moves towards the equator; in boreal winter, when the ITCZ is farthest south, if the insolation is higher in both the northern and southern hemispheres, the rainbelt is located further

north (Singarayer et al., 2017). But, on the one hand, this view still need more research to support; on the other hand, we are not sure whether the ITCZ was located at its northern/southern extremes during the early HS1.

Recently, mounting evidence suggests that the precipitation anomaly in the tropical west Pacific and Indian Ocean has a strong correlation with EI Niño-Southern Oscillation (ENSO) activities (e.g. Thirumalai et al., 2019). We revised our statement and propose that **the low global temperature, warming SST in tropical Indian Ocean and ENSO activities are the driving factors for the contraction of tropical precipitation belt in the Asian Monsoon region (Line 191-216)**. During the early HS1, the global mean temperature was very low, and the tropical SST warming developed rapidly in the low latitude of the Indian Ocean, which may lead to reduced range of latitudinal movement of the tropical rain belt. At the same time, the low temperature gradient between West and East tropical Pacific indicate that there was a EI Niño-like condition at this interval (Zhang et al., 2022, *NC*; Koutavas and Joanides, 2012, *Paleoceanography*). And recent work indicates that the range of Hadley circulation would contracte equatorward and become weak under EI Niño condition (Wodzicki and Rapp, 2020, *JC*; Guo and Li, *Advances in Climate Change Rearch*).

13. In conclusion, our findings highlight the presence of dry hydrological conditions in both the northern and southern parts of the Asian-Indian Monsoon region during the Early HS1. These dry conditions cannot be explained by a simple southward drift of the tropical rain-belt. Analyzing paleo-records from the northern low latitudes in the Asian-Indian Monsoon region, we established that a narrow tropical rain-belt existed during the Early HS1. This study presents a clear picture of the variations in hydrological conditions during the Early HS1 in the Asian-Indian Monsoon region. The last sentence might be replaced by a summary of the mechanisms driving the narrowed ITZC from both Northern- and Southern-Hemisphere perspectives. This would make for a considerably stronger conclusion statement and enhance the relevance to the wider scientific community targeted by Nature Communications.

[Response]

Thanks very much for your constructive suggestion.

We have modified the last paragraph following your suggestion, please see **Line 217-225**.

Reviewer #2 (Remarks to the Author):

Yang and others focus on the response of the tropical Intertropical Convergence Zone (ITCZ) feature over the Indian Ocean during Heinrich Stadial 1 (HS1) by presenting a new record of seawater- $\delta^{18}\text{O}$ (a proxy for salinity; $\delta^{18}\text{O}_{\text{sw}}$) from foraminiferal geochemistry in marine sediments proximal to the Ninetyeast Ridge and by compiling a series of previously published records. Their record shows lower $\delta^{18}\text{O}_{\text{sw}}$ (or lower surface-salinity) during the initial part of HS1 and Yang and colleagues suggest that this is in accordance with a series of other records showing intensified rainfall offshore the tip of peninsular India & in the northern western Pacific Warm Pool region. These records are in contrast to widespread dual-hemispheric and pan-Indian Ocean records of drought conditions during HS1. Thus, the authors argue that the ITCZ was restricted to a narrow band during the early part of HS1 and that rainfall in this contracted ITCZ was more intense. Whereas the premise is interesting and their record will certainly be of value to the paleoceanographic community, I am unable to recommend this manuscript for publication in Nature Communications as I am yet unconvinced by their mechanistic arguments. I detail my points below:

Thank you for your critical and valuable comments. We have revised the manuscript accordingly, and hoped the revised manuscript can be of satisfactory to the reviewer. Our point-by-point responses are presented below.

The questions about mechanism above have been answered in detail in **Comment #6** and **Comment #7**.

1. Proxy uncertainties and evidence for a “wetter” yet narrower ITCZ band during the early part of HS1: I am not convinced of the purported “wetter” records that are presented in the manuscript (Fig. 3 & Table S1). For one, it is extremely challenging to reconstruct precipitation using surface-ocean $\delta^{18}\text{O}$ and at the very least, the authors must propagate the uncertainties appropriately from paired Mg/Ca- $\delta^{18}\text{O}$ data (see e.g. Rohling, 2000; Thirumalai et al. 2016; Gray & Evans, 2019 etc.). Thus, it is unclear whether the changes shown in Fig. 3 are actually consistent with less saline conditions during HS1, given that anomalies are on the order of $\sim 0.1\text{--}0.3\text{‰}$ across all records. Please perform a robust error propagation exercise to see whether such an anomaly persists.

[Response]

Thank you very much for your useful suggestions.

We have performed a robust error propagation exercise to see whether such an anomaly persists in Methods part (**Line 262-276**) in revised manuscript. We also fully acknowledged the difficulty in reconstructing precipitation using surface-ocean $\delta^{18}\text{O}$ in the main text, and added more justification for our method in supplementary (**supplementary Line 56-81**) .

Firstly, the measured $\delta^{18}\text{O}_{\text{ruber}}$ values in Core I106 show a decline of $\sim 0.68\text{‰}$ at 16.3-18.3 ka, and this. As we know that $\delta^{18}\text{O}_{\text{ruber}}$ change is mainly controlled by seawater temperature and salinity. Meanwhile, Mg/Ca-SST in Core I106 appear shows about $0.65\text{ }^{\circ}\text{C}$ increase at 16.3-18.3 ka, which we believe inferring can explain about 0.15‰ decrease of $\delta^{18}\text{O}_{\text{ruber}}$ ($\delta^{18}\text{O}$ 0.23‰ change per $1\text{ }^{\circ}\text{C}$). We therefore assume that the about 0.53‰ decrease of $\delta^{18}\text{O}_{\text{ruber}}$ value in Core I106 was is mainly associated with seawater salinity (**Line 73-77**).

Secondly, as your suggested, we checked the $\delta^{18}\text{O}_{\text{sw}}$ errors of all records in Figure 3. We estimated the errors for Mg/Ca-SST and $\delta^{18}\text{O}_{\text{sw}}$ in cores I106, ADM-159 by propagating the uncertainties appropriately from paired Mg/Ca- $\delta^{18}\text{O}$ data using equations from Mohtadi et al. (2014), and we collected the $\delta^{18}\text{O}_{\text{sw}}$ errors from the original papers for other records (please see following table).

Core	Conditions in early HS1	SST error estimation	$\delta^{18}\text{O}_{\text{sw}}$ error estimation:	Methods
I106	wet	$\pm 1.03^{\circ}\text{C}$	$\pm 0.23\text{‰}$	Mohtadi et al., 2014
ADM-159	wet	$\pm 1.54^{\circ}\text{C}$	$\pm 0.27\text{‰}$	Mohtadi et al., 2014
SK129/CR04	wet	$\pm 0.7^{\circ}\text{C}$	$\pm 1\text{psu}$	Rohling, 2007
MD97-2141	wet	--	--	--
SO189-39KL	dry	$\pm 1.00^{\circ}\text{C}$	$\pm 0.30\text{‰}$	Mohtadi et al., 2014
GeoB10069-3	dry	--	$\pm 0.30\text{‰}$	Gibbons et al., 2014

Our new analysis suggests $\delta^{18}\text{O}_{\text{sw}}$ in Core I106 declines 0.51‰ at 16.3-18.3 ka, which is greater than the propagating error value (0.23‰). For Core MD97-2141, we cannot get its Mg/Ca analysis error and/or equation's error, hence it is difficult for us to estimate the propagate uncertainties. But its $\delta^{18}\text{O}_{\text{sw}}$ value at 16.4-18.3 ka is about 0.20‰ higher than it at 14.6-16.4 ka. SK129/CR04 show about 0.5 psu decline at 16.3-19.5 ka. For Core ADM-159, $\delta^{18}\text{O}_{\text{sw}}$ values at 16.5-18.7 ka show about 0.15‰ higher than that at 14.7-16.3 ka. $\delta^{18}\text{O}_{\text{sw}}$ values in Core SO189-39KL exist about 0.28‰ increase

during 14.8-18.6 ka. $\delta^{18}\text{O}_{\text{sw}}$ values in Core GeoB10069-3 exist about 0.3‰ increase at 15.3-19.5 ka.

Thirdly, in addition to these $\delta^{18}\text{O}_{\text{sw}}$ records, other geochemical proxies also recorded the anomalies in this region. Grain-size, dry bulk density and Si/Al in Core CG2 from the southern South China Sea illustrate that strong precipitation during 19.0-18.0 ka BP and 17.5-16.1 ka BP (Huang et al., 2019). XRF-derived $\log(\text{Fe}/\text{Ca})$ records in Core MD06-3075, a robust proxy for runoff-driven sedimentary discharge from Mindanao, show high precipitation occurred at 15.7-17.9 ka, and decreased precipitation at 15.0-15.7 ka (Fraser et al., 2014). And Fraser et al. (2014) have pointed out that an increased precipitation at Mindanao and dry conditions in Borneo and China at 15.7-17.8 ka (Fraser et al., 2014). Besides, Kaolinite content in Core I106, which is considered to be from the Sumatra, decreased a lot at about 16.8-20.4 ka, also implying dry hydrological environment in the Sumatra (Xu et al., 2022). Taking these evidence and consideration together, it is plausible to conclude that there was a wet hydrological environment at the north of Equator, and dry condition at north and south hemispheres.

Indeed, we cannot totally exclude the possibility that the anomaly recorded by $\delta^{18}\text{O}_{\text{sw}}$ proxy is affected by other factors. in the light of the results of other proxies' records in tropical Pacific and Indian Ocean, and the paleoclimatic records results in the Africa (Collins et al., 2010; Otto-Bliesner et al., 2014). Still, we feel that our analyses can support the alternative hypothesis that the tropical circulation system weakened and the tropical precipitation belt shrank during the early HS1. Nevertheless, we have fully discussed this uncertainty in the discussion section (Line 108-125), and hope our finding can motivate new studies on rethinking the simple south-north migration mechanism on tropical rain belt during North Atlantic cold events.

2. Moreover, the authors suggest that they only use the Dekens et al. 2002 equation for inverting SSTs - they must additionally test for the influence of salinity (e.g. see Gray & Evans, 2019) on Mg/Ca and see whether their result holds. They should perform the same analysis across all purported records consistent with “fresher” conditions.

[Response]

Thank you very much for your constructive suggestions!

We fully agree with your comment that seawater salinity could have impacts on planktonic foraminiferal Mg/Ca. In the light of your suggestion, we downloaded Paleo-Seawater Uncertainty Solver (PSU) toolkit (Thirumalai et al. 2016), and re-evaluate Mg/Ca and $\delta^{18}\text{O}_{\text{sw}}$ results across all purported records in this study (please see **supplementary, Line 56-80**).

We evaluated the influence of salinity on Mg/Ca variability of cores I106, ADM-159, MD97-2141 and SK129/CR04 by PSU Solver with two sigma analytical uncertainty of Mg/Ca and $\delta^{18}\text{O}$ and $\pm 5\%$ age uncertainty. The recalculated Mg/Ca-temperature results by PSU Solver in cores I106, ADM-159, MD97-2141 and SK129/CR04 are consistent with their published data of these studies (please see **supplementary Fig. 3**). The potential effects of salinity on these Mg/Ca-temperature results are very minor in these studies.

Moreover, we recalculated $\delta^{18}\text{O}_{\text{sw}}$ results by PSU Solver for cores ADM-159, MD97-2141 and SK129/CR04. The difference between original $\delta^{18}\text{O}_{\text{sw}}$ values and $\delta^{18}\text{O}_{\text{sw}}$ values recalculated by PSU Solver in Core SK129/CR04 is relatively big, because there is not $\delta^{18}\text{O}_{\text{sw}}$ equation of Epstein et al. (1953) in PSU Solver, so we recalculated $\delta^{18}\text{O}_{\text{sw}}$ values of Core SK129/CR04 by equation of Bemis et al (1998) in the PSU. **Overall, the changing trends of $\delta^{18}\text{O}_{\text{sw}}$ values in cores I106, ADM-159, MD97-2141 and SK129/CR04 are also consistent with results using their original dates. And their PSU Solver $\delta^{18}\text{O}_{\text{sw}}$ all also exist a negative abnormal event in the early Heinrich Stadial I.** Therefore, the Mg/Ca-SST and $\delta^{18}\text{O}_{\text{sw}}$ in these cores lead to robust reconstruction the paleo-climate in this region.

3.Finally, the Sulu Sea & Mindanao Dome records seem to show a steady increase in $\delta^{18}\text{O}_{\text{sw}}$ over the early part of HS1 and contrast with the record from Core I106; yet this discrepancy is not explained.

[Response]

Thanks for your comment.

The $\delta^{18}\text{O}_{\text{sw}}$ values in Core MD97-2141 from the Sulu Sea and Core I106 from this study both show a decrease during the early part of HS1 (please see the figure below). In the Sulu Sea, $\delta^{18}\text{O}_{\text{sw}}$ value

from Core MD97-2141 (Rosenthal et al., 2003) is about 1.19‰ at 14.6-16.4 ka, and then decreased to 1.00 ‰ at 16.3-18.3 ka, and the $\delta^{18}\text{O}_{\text{sw}}$ values continued to decline after 18.3 ka (Fig. R1). For Core I106, $\delta^{18}\text{O}_{\text{sw}}$ value is about 0.25‰ at about 15-16 ka, and then decreased abruptly to -0.08 ‰ at about 16.3-18.3 ka, and increased to 0.42‰ at 18.6-19.5 ka. After 19.5 ka, $\delta^{18}\text{O}_{\text{sw}}$ value also declined gradually. **The $\delta^{18}\text{O}_{\text{sw}}$ value in Core MD97-2141 and Core I106 both show that $\delta^{18}\text{O}_{\text{sw}}$ value declined substantially at about 16.5-18.3 ka (early HS1), indicating the hydrological condition in the early HS1 was wetter than that in the late HS1 (Fig. 3).** In addition, XRF-derived log (Fe/Ca) records from MD06-3075 (6°29'N, 125°50'E) at Mindanao, a robust proxy for freshwater runoff, document increased precipitation at Mindanao at 15.7-17.8 ka, and drier conditions in Borneo and China at this interval (Fraser et al., 2014) (please see **supplementary Fig. 4F**). (Line 130-135)

Fig. R1 $\delta^{18}\text{O}_{\text{sw}}$ records of cores I106 and MD97-2141

4. Moreover, I do not know if the average $\delta^{18}\text{O}_{\text{sw}}$ from the LGM to the early part of HS1 in these records (Rosenthal et al. 2003; Boillet et al. 2011) are even significantly different from one another. This needs to be tested within uncertainty.

[Response]

Thanks for your comment.

On the one hand, some age points in Core MD06-3067 (Boillet et al., 2011) from Mindanao Dome are based on $\delta^{18}\text{O}$ event instead of AMS ^{14}C measured on planktonic foraminifera. And the revised age model based on AMS ^{14}C dates for Core MD06-3067 with Marine 20 curve is quite different

from the original age model. **Thus, we removed the $\delta^{18}\text{O}_{\text{sw}}$ dates of Core MD06-3067 in this study.**

XRF-derived log (Fe/Ca) records from MD06-3075 (6°29'N, 125°50'E) at Mindanao, a robust proxy for freshwater runoff, documented increased precipitation at Mindanao at 15.7-17.8 ka (early HS1) (please see supplementary Fig. 4F), but with drier conditions in Borneo and China at this interval (Fraser et al., 2014).

On the other hand, we have performed validity test on the Mg/Ca and $\delta^{18}\text{O}_{\text{sw}}$ for MD97-2141 by PSU Solver (please see supplementary Fig. 3), the results show that the **changing trends of $\delta^{18}\text{O}_{\text{sw}}$ values in Core MD97-2141 is consistent with original date, and also exist a humid condition during the early Heinrich stadial 1.**

5. Winter Monsoon Rainfall: Thus, it appears that only the I106 record seems to show a structured anomalous response of lower $\delta^{18}\text{O}_{\text{sw}}$ in the sub-selected 7 records purported to show higher rainfall during the early part of HS1 (although the robustness is to be tested as indicated above). Whereas the Sulu Sea & Mindanao records response are uncertain, in the Indian Ocean side, the CR04 (Fig. 3D) also seems to be uncertain and not robust in its response. This leaves potentially the SS3827G record - which is only a $\delta^{18}\text{O}$ calcite record without a temperature measurement. Thus it is also uncertain in its hydroclimatic response. Nevertheless, if this lower $\delta^{18}\text{O}$ -calcite was indeed caused by lower $\delta^{18}\text{O}_{\text{sw}}$, then the authors need to address the idea that this is due to the advection of fresh runoff from the southeastern Indian coast due to strengthened winter monsoon rainfall (see e.g. Sarkar et al. 1990 or Kumar & Ramesh, 2017). Is it possible that these low salinity waters also made their way towards the core site? That said, the authors do not provide enough clarity to rule out that these different core sites might be representing rainfall/salinity during different parts of the year. This dimension needs to be discussed in a revised manuscript.

[Response]

Thank you very much for your comment.

1) the robustness for these selected records has been tested in revised manuscript (Please see supplementary). we have re-evaluated the Mg/Ca-SST and $\delta^{18}\text{O}_{\text{sw}}$ records in cores I106, ADM-159,

MD97-2141 and SK129/CR04 using PSU Solver with age uncertainty and analytical uncertainty. Recalculated Mg/Ca-SST and $\delta^{18}\text{O}_{\text{sw}}$ results are consistent with the original results. (Please see supplementary, Fig. 3).

2) we have added discussion on the effect of winter monsoon in our revised manuscript (Line 109-124; Line 127-138). The reconstructed $\delta^{18}\text{O}_{\text{sw-residual}}$ and SSS values from Core SK129/CR04 (6°29.65'N, 75°58.68'E) in the Equatorial Arabian Sea exhibit a low salinity event at 19.5-16.5 ka BP (Mahesh et al., 2014). And $\delta^{18}\text{O}$ records in multiple planktonic foraminiferal species from the Equatorial Arabian Sea also show a negative peak at about 19.0-17.0 ka BP (Sarkar et al., 1990, Tiwari et al., 2005). These “low salinity event” from cores SK129/CR04, SS3827G, SK-20-185 and SK-20-186 at the junction of two sea basins have been considered to be related with the strengthen of low-salinity water from the Bay of Bengal into the East Arabian Sea due to a stronger winter monsoon current (Sarkar et al., 1990, Tiwari et al., 2005).

However, Govil and Naidu (2010) have proposed that enriched $\delta^{18}\text{O}_{\text{sw}}$ values around 20 ka and 17 ka in cores SK17 and AAS9/21 from the eastern Arabian Sea represent less flow from the Bay of Bengal due to weaker winter monsoon. $\delta^{18}\text{O}_{\text{sw}}$ values in Core SK218/1 from the western BOB, which was affected by East Indian Coast Current, increased a lot during the HS1, also suggesting the weak winter monsoon current (Govil and Naidu, 2011) (Figure 1). These records are contrast to the idea that the strengthening of low-salinity water from the Bay of Bengal into the East Arabian Sea due to the stronger winter monsoon currents (Sarkar et al., 1990, Tiwari et al., 2005). Simultaneously, modern observation shows that there is an anti-clockwise winter current in the Bay of the Bengal (Please see Figure 1 in the revised manuscript), and less fresh water from the north of Bay of the Bengal would be injected into our location. The seawater in south of Bay of the Bengal is saltier than it in our study area. And more salty seawater would be transport into our study area by Winter Monsoon Currents if the winter monsoon became strengthen. But the $\delta^{18}\text{O}_{\text{sw}}$ in Core I106 became negative remarkably in early HS1. In addition, the seawater upwelling in northeast Indian Ocean would also be enhanced if the winter monsoon became stronger during 19-17 ka as recorded in cores SS3827G (Tiwari et al., 2005), SK-20-185 and SK-20-186 (Sarkar et al., 1990), and more salty deep water would be brought to the surface, which would give rise to the increase of $\delta^{18}\text{O}_{\text{sw}}$ values in cores I106 and ADM-159 from East Indian Ocean instead of decrease.

In general, we cannot rule out the possible effects of the tropical precipitation on “low salinity event” in Cores SS3827G, SK-20-185 and SK-20-186, because the negative *G. ruber* $\delta^{18}\text{O}$ and $\delta^{18}\text{O}_{\text{sw}}$ records in the Early HS1 in Core I106, and the increase in $\Delta\delta^{18}\text{O}_{\text{ruber-dutertrei}}$ from Core 758 (5°23.5'N, 90°21.67'E) from East Indian Ocean suggested strengthened ocean stratification and weakened wind during the HS1. Therefore, we have fully discussed these potential processes in the main text that still propose that increased tropical precipitation over the ocean plays a role in the SSS decline during the early HS1 in the northern Equator (**Line 109-124; Line 127-138**).

3) The $\delta^{18}\text{O}_{\text{sw}}$ value at 16.4-18.3 ka was obviously lower than it at 14.6 -16.4 ka in Core MD97-2141 in Sulu Sea (Rosenthal et al., 2003), indicating that surface seawater became fresher during the early HS1 than it during the late HS1, which was possibly associated with tropical precipitation. XRF-derived log (Fe/Ca) records from MD06-3075 (6°29'N, 125°50'E) at Mindanao, a robust proxy for freshwater runoff, also document that an increased precipitation at Mindanao at 15.7-17.8 ka (please see **supplementary Fig. 4F**), but with drier conditions in Borneo and China during this interval (Fraser et al., 2014).

6. Forcing Mechanisms: The proposed mechanisms for the interaction of orbital forcing with abrupt climate change do not take into account the latest results suggesting that orbital forcing modulates millennial-scale activity. That said, I was confused while reading the mechanism because the authors do not explain what happens to the ITCZ in the southern hemisphere of the tropical Indian Ocean, which when DJF insolation is relatively higher, should exhibit a seasonally more southern ITCZ in the Southern Hemisphere. How does this affect their proxy comparison?

[Response]

Thanks for your comment. This issue is also raised by reviewer#1.

1) Based on climatic model results, Singarayer et al. (2017) have proposed that there was ocean dominated expansion and contraction of the tropical rainbelt during the late Quaternary. This expansion/contraction is the result of the different response of the marine ITCZ when at its northern and southern extremes. In boreal summer, when ITCZ is farthest north, if the insolation is the lowest, the ITCZ moves towards the equator; in boreal winter, when the ITCZ is farthest south, if the insolation is higher in both the northern and southern hemispheres, the rainbelt is located further

north (Singarayer et al., 2017). But, on the one hand, this view still need more research to support; on the other hand, we are not sure whether the ITCZ was located at its northern/southern extremes during the early HS1.

2) Mounting evidence suggests that the precipitation anomaly in the tropical west Pacific and Indian Ocean has a strong correlation with EI Niño-Southern Oscillation (ENSO) activities (e.g. Thirumalai et al., 2019). We revised our statement and propose that **the low global temperature, warming SST in tropical Indian Ocean and ENSO activities are the driving factors for the contraction of tropical precipitation belt in the Asian Monsoon region (Line 180-204)**. During the early HS1, the global mean temperature was very low, and the tropical SST warming developed rapidly in the low latitude of the Indian Ocean, which may lead to reduced range of latitudinal movement of the tropical rain belt. At the same time, the low temperature gradient between West and East tropical Pacific indicate that there was a EI Niño-like condition during the early phase of HS1 (Zhang et al., 2022, *NC*; Koutavas and Joannides, 2012, *Paleoceanography*). And recent work indicates that the range of Hadley circulation would contract equatorward and become weak under EI Niño condition, because that tropical troposphere become warmer, and subtropical troposphere is cooler, which enhances the meridional temperature gradient (MTG), and then results in shrinking of the Hadley circulation in both Hemispheres under EI Niño-like state (Wodzicki and Rapp, 2020, *JC*; Guo and Li, *Advances in Climate Change Research*).

In all, the cooling in northern high latitudes prevented the northward expanding of the Hadley circulation, as suggested by dry condition records in the Northern Hemisphere during the early HS1, and strong EI Niño activities also lead to shrinking of the Hadley circulation extent in the Southern Hemisphere (**Line 217-225**).

7. Comparisons with models: The authors do not present comparisons with climate model output - which by itself is not a problem. However, there are several papers (e.g. Mohtadi et al. 2014) which use model simulations and show that there is evidence for ITCZ movement but not intensification - let alone intensification associated with a contraction. Can the authors speculate as to why this may be the case? Can they rule out that the seasonal-bias of different proxies affects this finding and that it can be applied to mean-annual ITCZ shifts?

[Response]

Thank you very much for your comment.

In the revised manuscript we discussed some relevant model results including Mohtadi et al. (2014) from the Indian Ocean Summer Monsoon region for comparison (**Line170-190**). We notice that some model results suggest that the ITCZ migrated southward in response to the reduction of AMOC during the HS1, which are not in agreement with our hypothesis. We have fully discussed this issue in the revised manuscript, detailed below.

Firstly, due to the limitation in experimental set-up, **previous model simulation is unable to investigate the tropical precipitation pattern during the Heinrich stadial I in two separated stages**. Model results from Mohtadi et al. (2014) indicate that “a negative rainfall anomaly over Sumatra, whereas rainfall over southern Indonesia increase during the HS1”. We also did similar analysis for TRACE21k data (Liu et al., 2009; 2014), and the simulated rainfall response seems to be similar (Fig. R2). There is indeed contradiction between model simulations and proxy data. The paleoclimate records in VM33-80 from south Indonesia (also cited by Mohtadi et al., 2014) show humid condition at 16.9-14.7 ka, and its hydrological condition was dry in the early HS1, instead of humid condition during the whole HS1 (about 19-15 ka) (Muller et al., 2012). And $\delta^{18}\text{O}_{\text{sw}}$ records in MD98-2165 (Levi et al., 2007), MD01-2378 (Xu et al., 2008), GeoB10069-3 (Gibbons et al., 2014), stalagmite $\delta^{18}\text{O}$ record from Ball Gown (Denniston et al., 2013) all suggest regional drier conditions in the early HS1. Reconstructed precipitation records in Southeast Africa also show humid condition around 16.9 ka-14.7 ka, instead of in the early HS1 (Schefuß et al., 2011) (be referenced by Mohtadi et al., 2014). Although some records in Southern Africa show wetter condition during the HS1, highlighting complicated spatial pattern, but on the whole, there are more arid records than humid records (Thomas et al., 2012).

In the past, there was lack of paleoclimate records in the north of the Equator in the Indian Ocean to identify a credible two-phase tropical hydroclimate variation there. But some recent studies proposed that there is a two-phase tropical hydroclimate in tropical Asian region, which seems to be in line with the two-stage variation of AMOC. The south-north migration pattern of the ITCZ is a too simple hypothesis to explain the more complicated interhemispheric synchronous drought during the early HS1, and we agree that the contraction of the tropical rain-belt was possibly affected

by multiple processes, such as low global temperature, tropical warming and El Niño activities related to the climatic change in the northern high latitudes, rather than a simple north-south migration mechanism.

Secondly, some proxy-records and model results also have suggested that there was hemispherically symmetric contraction of the tropical rain-belt in response to the cold events of the North Atlantic. For example, model simulations from Africa have also revealed that precipitation coherency decreased in both Southeastern Equatorial and Northern Africa in response to meltwater-induced reductions in the AMOC during the early phases of the last deglaciation (Otto-Bliesner et al., 2014). Collins et al. (2010) proposed that the tropical rain-belt in Africa contracted relative to the Late Holocene during the HS1, owing to a latitudinal compression of atmospheric circulation related to a lower mean global temperature. Yan et al. (2015) also pointed out that the latitudinal range of ITCZ rainfall in the Western Pacific contracted over decadal to centennial timescales in response to a cold climate during the Little Ice Age (LIA). Stalagmite record from southwest Madagascar also discover that tropical rain-belt simultaneously expands or contracts in both hemispheres in the past (Burns et al., 2020).

Thirdly, although we cannot totally rule out that the seasonal-bias of different proxies affects this finding, we would argue that the $\delta^{18}\text{O}$ and Mg/Ca values reconstructed from about 80 *G. ruber* shells in Core I106 in this study could be taken as annual signals (randomly distributed throughout a year), rather than as indicative of seasonal changes. Based on our sediment trap data, modern planktonic foraminifera *G. ruber* exists all year round in our study area.

Fig. R2 Annual precipitation difference between the LGM (21ka) and HS1 (17ka) (500-year mean for each period) in TRACE 21k simulation (labelled “ALL”) and four single forcing TRACE simulations (labelled “ORB”, “CO2”, “ICE”, “MWF” for orbital forcing, greenhouse gas forcing, ice-sheet forcing and melt-water flux forcing).

Our deepest gratitude goes to reviewers for reviewers’ careful work and thoughtful suggestions that have helped improve this paper substantially.

References:

- Gray, W. R., & Evans, D. (2019). Nonthermal Influences on Mg/Ca in Planktonic Foraminifera: A Review of Culture Studies and Application to the Last Glacial Maximum. *Paleoceanography and Paleoclimatology*, 34(3), 306–315. <https://doi.org/10.1029/2018pa003517>
- Kumar, P. K., & Ramesh, R. (2017). Revisiting reconstructed Indian monsoon rainfall variations during the last ~25ka from planktonic foraminiferal $\delta^{18}\text{O}$ from the Eastern Arabian Sea. *Quaternary*

International, 443, 29–38. <https://doi.org/10.1016/j.quaint.2016.07.012>

Mohtadi, M., Prange, M., Oppo, D. W., De Pol-Holz, R., Merkel, U., Zhang, X., Steinke, S., & Lückge, A. (2014). North Atlantic forcing of tropical Indian Ocean climate. *Nature*, 509(7498), 76–80. <https://doi.org/10.1038/nature13196>

Rohling, E. J. (2000). Paleosalinity: confidence limits and future applications. *Marine Geology*, 163(1–4), 1–11. [https://doi.org/10.1016/s0025-3227\(99\)00097-3](https://doi.org/10.1016/s0025-3227(99)00097-3)

Rosenthal, Y., Oppo, D. W., & Linsley, B. K. (2003). The amplitude and phasing of climate change during the last deglaciation in the Sulu Sea, western equatorial Pacific. *Geophysical Research Letters*, 30(8), 1428–1424. <https://doi.org/10.1029/2002GL016612>

Sarkar, A., Ramesh, R., Bhattacharya, S. K., & Rajagopalan, G. (1990). Oxygen isotope evidence for a stronger winter monsoon current during the last glaciation. *Nature*, 343(6258), 549-551. <https://doi.org/10.1038/343549a0>

Thirumalai, K., Quinn, T. M., & Marino, G. (2016). Constraining past seawater $\delta^{18}\text{O}$ and temperature records developed from foraminiferal geochemistry. *Paleoceanography*. <https://doi.org/10.1002/2016PA002970>

Reviewer #1 (Remarks to the Author):

Yang 2022 ITCZ HS1 Rev #2

The authors have adequately addressed the questions/concerns raised in my initial review of the manuscript. The synthesis supports the contention of an unresolved question regarding climatic state during HS1 and the data presented and interpretation thereof are internally consistent, providing an explanation consistent with both the land and ocean evidence for the HS1 interval of time. The proposed mechanisms are broadly testable in climate-model space and sufficiently provocative to motivate that community.

The manuscript will require substantial editing for grammar, some of which is address below, along with other editorial, scientific comments and clarifications.

13 Despite the responses of the tropical hydroclimate to a North Atlantic cooling event during the Heinrich
14 Stadial 1 (HS1) have been extensively studied in African, South American and Indonesian; the nature of
15 such responses remains many debates

Despite the fact that the response of the tropical hydroclimate to a North Atlantic cooling event during the Heinrich Stadial 1 (HS1) has been extensively studied in African, South American and Indonesian; the nature of such responses remains debated.

This

21 study reveals that low global temperatures and El Niño activates both exerted a profound influence on the
22 tropical hydroclimate in the Indo-Asian-Australian monsoon region during the early HS1.

This study reveals that low global temperatures and El Niño both exerted a profound influence on the tropical hydroclimate in the Indo-Asian-Australian monsoon region during the early HS1.

We assume that precipitation in our study area was also linked with
52 IOM and ITCZ rain belt system during HS1 because that marine-continental tectonic is consistent with it
53 today and the structure of land-sea thermodynamic between Indian Ocean and Eurasian Continent also
54 exist.

The land-sea continent configuration argument for justification of the assumption is insufficient; a great many factors can come to play in this regard, the least of which is tectonics, at the time-scale of this work. Presenting this argument will garner far more controversy than simply stating the assumption and carrying on.

67 Our plankton tow samples from the study area show that *G. ruber* is mainly distributed in water
68 depths of 0-50 m, and that it can therefore be classed as a surface species 18

A mixed-layer species?

79 The observed salinity and $\delta^{18}\text{O}_{\text{sw}}$ values of surface water in the Northeastern Indian Ocean reported

80 by Gebregiorgis et al.²¹ prove that $\delta^{18}\text{O}_{\text{sw}}$ values show a linear correlation with salinity in our study area.

The reference is for the coastal Andaman Islands, Andaman Sea region, not the equatorial region of the IO. Although very likely linear in this region as well, the $\delta^{18}\text{O}_{\text{sw}}$ - salinity relationship cited does not prove the correlation in the study area.

Caley et al., 2015 (10.1002/2014PA002720) and references therein may provide a better reference?

255 Mg/Ca-SST and $\delta^{18}\text{O}_{\text{sw}}$ reconstruction. Mg/Ca values were converted to temperature using the equations developed

256 by Anand et al. (2003) **71**: $\text{Mg/Ca} [\text{mmol mol}^{-1}] = 0.38e^{0.09T[^\circ\text{C}]}$. Seawater $\delta^{18}\text{O}$ ($\delta^{18}\text{O}_{\text{sw}}$) values were calculated using the

257 equation proposed by Bemis et al. (1998) **20**: $T [^\circ\text{C}] = 14.9 - 4.8 (\delta^{18}\text{O}_{\text{c}} - \delta^{18}\text{O}_{\text{sw}})$. An additional 0.27‰ was added to them

258 to convert the Vienna Pee Belemnite (VPDB) values to Vienna Standard Mean Ocean Water (VSMOW) values. $\delta^{18}\text{O}$

Please explicitly provide the $\delta^{18}\text{O}_{\text{sw}}$ -salinity relationship used

Meanwhile, there would be more salt water

131 transported from the south into our study area if the winter Monsoon Current strengthened, on the contrary,

$\delta^{18}\text{O}_{\text{sw}}$ values declined a lot in Core I106 during the early HS1.

Meanwhile, there would be more salt water transported from the south into our study area if the winter Monsoon Current strengthened; on the contrary, $\delta^{18}\text{O}_{\text{sw}}$ values declined a lot in Core I106 during the early HS1.

Tropical

165 hydroclimate within HS1 from the north of the equator in the IAA monsoon realm also exist two distinct

166 phases, with wet hydrological condition at about 18.3-16.3 ka and dry condition at ~16.3-14.7 ka, which

167 was in line with this two-step AMOC slowdown (Fig. 4E).

Tropical hydroclimate within HS1 from the north of the equator in the IAA monsoon realm also exhibit two distinct

phases, with wet hydrological condition at about 18.3-16.3 ka and dry condition at ~16.3-14.7 ka, which was in line with this two-step AMOC slowdown (Fig. 4E).

Accordingly,

180 variations in tropical precipitation patterns are not only affected by the interhemispheric temperature

181 difference, but also be associated with other driving factors

Accordingly, variations in tropical precipitation patterns are not only affected by the

interhemispheric temperature difference, but also associated with other driving factors

Reviewer #3 (Remarks to the Author):

I have read with great interests the manuscript by Yang and co-workers "A contracting Intertropical Convergence Zone during the Early Heinrich Stadial 1" that present new foraminiferal data from a sediment core located in the southern Bay of Bengal to investigate the response of the Indian-Asian-Australian monsoon region to global and regional changes during Heinrich stadial 1. The authors propose more complex developments of the Intertropical Convergence Zone (ITCZ) than previously documented/thought during Heinrich Stadial 1, for which the occurrence of an early phase of contracted ITCZ rain belt north of the equator is claimed. The topic is very interesting and worth of publication in Nature Communications, the data are of an overall good quality although their interpretation could be deeper and more robust, and the analysis of the complementing evidence could (and should) be improved. The palaeocenographic time series presented lack the Holocene section, leaving the reader with the question of how relevant the amplitude of the discussed palaeoclimate is signals with respect to the glacial-interglacial hydroclimate changes in the region.

My review centres on two aspects of the manuscript, namely the (overlooked) role of the monsoon winds on the hydrography of the southern Bay of Bengal and the relationship between the findings of Yang et al. and the global temperature changes.

Role of the winds versus precipitation. The authors claim that their seawater $\delta^{18}\text{O}$ reconstructions from the southern Bay of Bengal reflects changes in direct precipitation and runoff alone. Besides the doubts I have on the impact of the latter on the hydrography of the southern Bay of Bengal, this claim is contrasted with the NCEP/NCAR 40-year reanalysis project (Kalnay et al., 1996, Bulletin of the American Meteorological Society), which highlights the importance of the winds associated with the Indian monsoon system in that sector of the bay. Accordingly, I wonder what the response of the surface ocean seawater $\delta^{18}\text{O}$ would be to a change (e.g., a weakening) of the wind mixing of the upper ocean during HS1? This hypothesis needs to be considered, because a shallower (smaller volume) mixed layer associated to a stratified surface ocean would "maximise" the negative seawater $\delta^{18}\text{O}$ anomaly also in absence of the increase in direct precipitation proposed by Yang et al. to interpret their data. Low-resolution and not comparably well-dated evidence from Bolton et al. (2013, Quaternary Science Reviews) is discussed quite superficially (lines 120-121) in the manuscript and in the supplementary information (lines 43-45). At glacial terminations, weakening of the monsoon circulation causes stratification in the southern Bay of Bengal. I recommend adding some depth to this aspect of the study, as I think that interpretation is not as straightforward as claimed by the authors. Some sort of sensitivity test of the impact of the mixed layer depth on the seawater $\delta^{18}\text{O}$ keeping precipitation constant may help addressing this point. Also, I am unaware that there is upwelling in the southern Bay of Bengal as suggested by the authors (lines 120-121).

Relationship between Hadley cell circulation and global temperature. This is a key aspect of the manuscript and I do not think the authors use the latest, up to date, statistically robust reconstruction of global surface temperature to convincingly tackle this issue. The dataset of Osman et al. (2021, Nature) shows that during the early phase of Heinrich stadial 1 discussed by the authors global temperature begins to warm (approximately at 17 ka BP), so it was not cold as claimed by Yang et al. (lines 191-192) nor is a time of global climate stability. Such a change would occur right in the middle of the early phase of Heinrich stadial 1 targeted by the authors. How would their current interpretation change (or hold) considering this new evidence?

Minor comments

The manuscript would benefit from being read through by an English native speaker.

Response to Reviewers' comments

Reviewer #1 (Remarks to the Author):

The authors have adequately addressed the questions/concerns raised in my initial review of the manuscript. The synthesis supports the contention of an unresolved question regarding climatic state during HS1 and the data presented and interpretation thereof are internally consistent, providing an explanation consistent with both the land and ocean evidence for the HS1 interval of time. The proposed mechanisms are broadly testable in climate-model space and sufficiently provocative to motivate that community.

1. The manuscript will require substantial editing for grammar, some of which is address below, along with other editorial, scientific comments and clarifications.

[Response]

We are very sorry for the language problems in this manuscript and inconvenience they caused in your reading. In this revised manuscript, the language presentation was improved with the aid of the [Scientific English Rewriting](https://rewriting.ai/) website (<https://rewriting.ai/>), which specializes in providing scientific English writing services for the fields of oceanography and geology. We have then proofread the grammar and scientific expression of the whole manuscript to ensure that it meets the standards of this journal.

2. Despite the responses of the tropical hydroclimate to a North Atlantic cooling event during the Heinrich Stadial 1 (HS1) have been extensively studied in African, South American and Indonesian; the nature of such responses remains many debates.

Despite **the fact that the response** of the tropical hydroclimate to a North Atlantic cooling event during the Heinrich Stadial 1 (HS1) **has** been extensively studied in African, South American and Indonesian; the nature of such responses remains **debated**.

[Response]

Thank you so much for your carefully reading and the improvement of the readability of the statement there. We have revised these sentences following your suggestions in **Lines 18-20 (in Manuscript with revising marks)**.

3. This study reveals that low global temperatures and El Niño ~~activates~~ both exerted a profound influence on the tropical hydroclimate in the Indo-Asian-Australian monsoon region during the early HS1.

This study reveals that low global temperatures and El Niño both exerted a profound influence on the tropical hydroclimate in the Indo-Asian-Australian monsoon region during the early HS1.

[Response]

Thanks for your careful reading.

This sentence has been revised following your suggestion in **Lines 27-28**.

4. We assume that precipitation in our study area was also linked with IOM and ITCZ rain belt system during HS1 because that marine-continental tectonic is consistent with it today and the structure of land-sea thermodynamic between Indian Ocean and Eurasian Continent also exist. The land-sea continent configuration argument for justification of the assumption is insufficient; a great many factors can come to play in this regard, the least of which is tectonics, at the time-scale of this work. Presenting this argument will garner far more controversy than simply stating the assumption and carrying on.

[Response]

Thank you for underlining this deficiency.

We have modified this sentence and deleted the part of “and the structure of land-sea thermodynamic between Indian Ocean and Eurasian Continent also exist” (**Lines 74-76**).

Our model simulation has shown that precipitation in study area was also affected by Indian Ocean Summer Monsoon and ITCZ during the HS1 (**Fig. R2 in the first responses to reviewers' comments**), as well as Mohtadi et al. (2014) (**Fig. R1**). Thus, we feel it is plausible to assumed that precipitation in our study area was also linked with IOM and ITCZ rain belt system during HS1.

Fig. R1 Results from the CCSM3 simulations of Heinrich stadial 1 (from Mohtadi et al., 2014)

5. Our plankton tow samples from the study area show that *G. ruber* is mainly distributed in water depths of 0-50 m, and that it can therefore be classed as a surface species **A mixed-layer species?**

[Response]

Thank you very much for this suggestion.

We have corrected “a surface species” to “a mixed-layer species” in **Line 89**.

6. The observed salinity and $\delta^{18}\text{O}_{\text{sw}}$ values of surface water in the Northeastern Indian Ocean reported by Gebregiorgis et al. prove that $\delta^{18}\text{O}_{\text{sw}}$ values show a linear correlation with salinity in our study area.

The reference is for the coastal Andaman Islands, Andaman Sea region, not the equatorial region of the IO. Although very likely linear in this region as well, the $\delta^{18}\text{O}_{\text{sw}}$ - salinity relationship cited does not prove the correlation in the study area. Caley et al., 2015 (10.1002/2014PA002720) and references therein may provide a better reference?

[Response]

We are grateful for this suggestion.

In response to the concerns raised by the reviewer, we have incorporated additional modern observed $\delta^{18}\text{O}_{\text{sw}}$ and salinity data from Delaygue et al. (2001), Singh et al. (2010) and Achyuthan et al. (2013) to cover the southern region of the Bay of the Bengal and the equatorial East Indian

Ocean. These data have been included in the “**Supplementary information on modern $\delta^{18}\text{O}_{\text{sw}}$ and salinity**”. These $\delta^{18}\text{O}_{\text{sw}}$ and salinity data exhibit a linear correlation, as depicted in **Figure S2**.

We have made relevant modifications to the main text (**in Lines 103-110**) and the Supplementary Information (**in Lines 47-50, 68-72**).

The modern $\delta^{18}\text{O}_{\text{sw}}$ and salinity data from the southern Bay of the Bengal and the Andaman Sea region demonstrate a strong linear correlation (Figure S2). This implies that the reconstructed $\delta^{18}\text{O}_{\text{sw}}$ values for Core I106 are indicative of the regional sea surface salinity (SSS) signal.

7. Mg/Ca-SST and $\delta^{18}\text{O}_{\text{sw}}$ reconstruction. Mg/Ca values were converted to temperature using the equations developed by Anand et al. (2003) 71: $\text{Mg/Ca} [\text{mmol mol}^{-1}] = 0.38e^{0.09T[^\circ\text{C}]}$. Seawater $\delta^{18}\text{O}$ ($\delta^{18}\text{O}_{\text{sw}}$) values were calculated using the equation proposed by Bemis et al. (1998): $T [^\circ\text{C}] = 14.9 - 4.8 (\delta^{18}\text{O}_{\text{c}} - \delta^{18}\text{O}_{\text{sw}})$. An additional 0.27‰ was added to them to convert the Vienna Pee Belemnite (VPDB) values to Vienna Standard Mean Ocean Water (VSMOW) values.

Please explicitly provide the $\delta^{18}\text{O}_{\text{sw}}$ -salinity relationship used.

[Response]

Thank you very much for your comment.

We are very sorry for the confusion we caused.

1) Our study aimed to analyze the relationship between modern $\delta^{18}\text{O}_{\text{sw}}$ and salinity in the southern Bay of Bengal. We found a strong linear correlation between $\delta^{18}\text{O}_{\text{sw}}$ and salinity, indicating that $\delta^{18}\text{O}_{\text{sw}}$ in this region can be used as an indicator of changes in regional sea surface salinity.

2) In this study, we reconstructed $\delta^{18}\text{O}_{\text{sw}}$ by Mg/Ca-SST and $\delta^{18}\text{O}$ values of planktonic foraminifera *G. ruber* using the equation of Bemis et al. (1998): $T [^\circ\text{C}] = 14.9 - 4.8 (\delta^{18}\text{O}_{\text{c}} - \delta^{18}\text{O}_{\text{sw}})$. The reconstructed $\delta^{18}\text{O}_{\text{sw}}$ in Core I106 could be considered an indirect salinity proxy, as $\delta^{18}\text{O}_{\text{sw}}$ and salinity showed a strong correlation in our study area (Figure S2). Previous paleo-records from the East Indian Ocean and tropical West Pacific have also used $\delta^{18}\text{O}_{\text{sw}}$ to indicate changes in SSS (Mohtadi et al., 2010; 2014; Rashid et al., 2007, 2011; Gibbons et al., 2014). **We did not calculate and provide SSS values for Core I106 in this study, as we wanted to directly compare our $\delta^{18}\text{O}_{\text{sw}}$ records with previous studies. Therefore, we feel it is not necessary to provide the equation of the $\delta^{18}\text{O}_{\text{sw}}$ -salinity relationship in the Methods part because it was not used.**

3) if necessary, we understand the formula in Figure S2 can be used to convert the reconstructed $\delta^{18}\text{O}_{\text{sw}}$ into SSS.

8. Meanwhile, there would be more salt water transported from the south into our study area if the winter Monsoon Current strengthened, on the contrary, $\delta^{18}\text{O}_{\text{sw}}$ values declined a lot in Core I106 during the early HS1.

Meanwhile, there would be more salt water transported from the south into our study area if the winter Monsoon Current strengthened; on the contrary, $\delta^{18}\text{O}_{\text{sw}}$ values declined a lot in Core I106 during the early HS1.

[Response]

Thanks for your carefully reading.

We have corrected this sentence following your suggestion in **Lines 178-180**.

9. Tropical hydroclimate within HS1 from the north of the equator in the IAA monsoon realm also exist two distinct phases, with wet hydrological condition at about 18.3-16.3 ka and dry condition at ~16.3-14.7 ka, which was in line with this two-step AMOC slowdown (Fig. 4E).

Tropical hydroclimate within HS1 from the north of the equator in the IAA monsoon realm also **exhibit** two distinct phases, with wet hydrological condition at about 18.3-16.3 ka and dry condition at ~16.3-14.7 ka, which was in line with this two-step AMOC slowdown (Fig. 4E).

[Response]

Thank you very much for pointing out our writing shortcomings.

We have corrected this sentence following your suggestion in **Lines 248-250**.

10. Accordingly, variations in tropical precipitation patterns are not only affected by the interhemispheric temperature difference, but also **be** associated with other driving factors.

Accordingly, variations in tropical precipitation patterns are not only affected by the interhemispheric temperature difference, but also associated with other driving factors

[Response]

Thanks for your carefulness and patience.

We have corrected this sentence following your suggestion in **Lines 233-235**.

Reviewer #3 (Remarks to the Author):

I have read with great interests the manuscript by Yang and co-workers “A contracting Intertropical Convergence Zone during the Early Heinrich Stadial 1” that present new foraminiferal data from a sediment core located in the southern Bay of Bengal to investigate the response of the Indian-Asian-Australian monsoon region to global and regional changes during Heinrich stadial 1. The authors propose more complex developments of the Intertropical Convergence Zone (ITCZ) than previously documented/thought during Heinrich Stadial 1, for which the occurrence of an early phase of contracted ITCZ rain belt north of the equator is claimed. The topic is very interesting and worth of publication in Nature Communications, the data are of an overall good quality although their interpretation could be deeper and more robust, and the analysis of the complementing evidence could (and should) be improved. The palaeoclimatographic time series presented lack the Holocene section, leaving the reader with the question of how relevant the amplitude of the discussed palaeoclimate is signals with respect to the glacial-interglacial hydroclimate changes in the region.

My review centres on two aspects of the manuscript, namely the (overlooked) role of the monsoon winds on the hydrography of the southern Bay of Bengal and the relationship between the findings of Yang et al. and the global temperature changes.

Thank you for your precious comments and suggestions, which are all valuable and very helpful for revising and improving our paper. We have revised the manuscript accordingly, and our point-by-point responses are presented below.

1. Role of the winds versus precipitation. The authors claim that their seawater $\delta^{18}\text{O}$ reconstructions from the southern Bay of Bengal reflects changes in direct precipitation and runoff alone. Besides the doubts I have on the impact of the latter on the hydrography of the southern Bay of Bengal, this claim is contrasted with the NCEP/NCAR 40-year reanalysis project (Kalnay et al., 1996, Bulletin of the American Meteorological Society), which highlights the importance of the winds associated with the Indian monsoon system in that sector of the bay. Accordingly, I

wonder what the response of the surface ocean seawater $\delta^{18}\text{O}$ would be to a change (e.g., a weakening) of the wind mixing of the upper ocean during HS1? This hypothesis needs to be considered, because a shallower (smaller volume) mixed layer associated to a stratified surface ocean would “maximise” the negative seawater $\delta^{18}\text{O}$ anomaly also in absence of the increase in direct precipitation proposed by Yang et al. to interpret their data. Low-resolution and not comparably well-dated evidence from Bolton et al. (2013, Quaternary Science Reviews) is discussed quite superficially (lines 120-121) in the manuscript and in the supplementary information (lines 43-45). At glacial terminations, weakening of the monsoon circulation causes stratification in the southern Bay of Bengal. I recommend adding some depth to this aspect of the study, as I think that interpretation is not as straightforward as claimed by the authors. Some sort of sensitivity test of the impact of the mixed layer depth on the seawater $\delta^{18}\text{O}$ keeping precipitation constant may help addressing this point. Also, I am unaware that there is upwelling in the southern Bay of Bengal as suggested by the authors (lines 120-121).

[Response]

We are extremely grateful to reviewer for pointing out this problem.

Firstly, in response to your valuable suggestions, we have obtained the modern wind speed and precipitation data from the NCEP reanalysis project (1948-2022, <http://apdrc.soest.hawaii.edu/las/v6/dataset?catitem=16341>) and sea surface salinity (SSS) data from the Hadley analyses project (1948-2022, <http://apdrc.soest.hawaii.edu/las/v6/dataset?catitem=16341>). **These datasets (1948-2022) reveal that the seasonal cycle in SSS change within a year is closely associated with precipitation in the study area, rather than wind speed (Fig. S1/Fig. R2).** The lowest SSS values are observed from August to November, coinciding with the highest precipitation. **We have also calculated the precipitation minus evaporation (P-E) fluxes in study area (Fig. R2), which evidently suggests that the change in SSS in the study area is closely linked to the flux of fresher water.**

Fig R2 Mean monthly precipitation (blue triangle), mean monthly SSS (orange circle), mean monthly wind speed (grey bar) and fresher water flux (Precipitation-Evaporation, red diamond) in study area.

Secondly, we also have conducted Mg/Ca and $\delta^{18}\text{O}$ analyses on planktonic foraminifera *Neogloboquadrina dutertrei*, a thermocline species, in Core I106. The temperature difference between sea surface (*G. ruber*) and thermocline layer (*N. dutertrei*) ($\Delta T_{\text{ruber-dutertrei}}$) in Core I106 exhibited an increase trend since ~21ka with several millennial-scale fluctuations, indicating a relatively weak water stratification during this period. **Notably, $\Delta T_{\text{ruber-dutertrei}}$ in Core I106 did not exhibit significant changes at 16.3-18.3 ka (Fig. R3).** We therefore consider that the abrupt increase in $\delta^{18}\text{O}_{\text{sw}}$ in Core I106 at 16.3-18.3 ka was primarily related to precipitation rather than water stratification.

Fig. R3 The temperature difference between sea surface and thermocline water and *G. ruber* $\delta^{18}\text{O}$ values since 24 ka.

Thirdly, on the one hand, the $\delta^{18}\text{O}_{\text{sw}}$ records in Core RC12-344 (Fig. R3) and ADM-159 in the northeastern Indian Ocean indicate that SSS was high during the last deglaciation period, and then it began to decrease at the onset of the Early Holocene, and SSS increased during the Late Holocene. However, the gradual increase in $\Delta T_{\text{ruber-dutertrei}}$ in Core I106 since 21 ka, indicates a weakened Indian Ocean Summer Monsoon and water stratification. **The changing trend in the $\delta^{18}\text{O}_{\text{sw}}$ records from the northeastern Indian Ocean is inconsistent with that of $\Delta T_{\text{ruber-dutertrei}}$ in Core I106 (Fig. R3).** These findings suggest that precipitation may have played a more important role in the long-term changes in SSS in this region, rather than wind intensity.

On the other hand, $\delta^{18}\text{O}$ records obtained from planktonic foraminifera *G. ruber* in the northeast Indian Ocean have shown that the $\delta^{18}\text{O}$ value was the highest during the Early Holocene and gradually declined since about 5.0 ka due to the weakening of summer monsoon precipitation during the Holocene (Rashid et al., 2007, 2011, Govil et al., 2011, Mahtadi et al., 2014, Gebregiorgis et al., 2018). Our $\delta^{18}\text{O}$ record in Core I106 also shows a similar changing trend from

the Early Holocene to the late Holocene, suggesting the change in $\delta^{18}\text{O}$ record may be also closely related to precipitation.

2. Relationship between Hadley cell circulation and global temperature. This is a key aspect of the manuscript and I do not think the authors use the latest, up to date, statistically robust reconstruction of global surface temperature to convincingly tackle this issue. The dataset of Osman et al. (2021, Nature) shows that during the early phase of Heinrich stadial 1 discussed by the authors global temperature begins to warm (approximately at 17 ka BP), so it was not cold as claimed by Yang et al. (lines 191-192) not is a time of global climate stability. Such a change would occur right in the middle of the early phase of Heinrich stadial 1 targeted by the authors. How would their current interpretation change (or hold) considering this new evidence?

[Response]

Thank you very much for your comments.

1) **we have added the latest statistically robust reconstruction of global surface temperature (Osman et al., 2021) in Figure 4 in this revised version.** The modelled global surface temperature shows that the Earth was in a ubiquitously cold glacial state at 24-17 ka, and the temperatures during this period in the high northern ($>45^\circ\text{N}$) and southern ($>45^\circ\text{S}$) latitudes were both cooler than the preindustrial era, below -20°C (Osman et al., 2021). The reconstructed global surface temperature by Shakun et al. (2012) also documents that there was about 0.3°C of global warming before about 17.5 ka. They both suggest that the global surface temperature was still quite low in the early phase of the HS1 (about 18.3-16.3), although it began to warm slightly. The global surface temperature in the latter phase of the HS1 increased remarkably.

2) meanwhile, during the Early HS1, there was a sudden increase in heat advection into the low latitudes of the Indian Ocean due to the anomalous transportation of heat northward into the northern high latitudes and a more vigorous Indonesian Throughflow (ITF) associated with the expansion of the Indo-Pacific Warm Pool (IPWP) (Deckker et al., 2012). This is supported by a steep and abrupt rise with a magnitude of $> 1.0^\circ\text{C}$ at about 19.5-18.0 ka in the East Indian Ocean (Mohtadi et al., 2010, 2014, Saraswat et al., 2005), and warmer SSTs events around 20 ka and 17 ka from northern Arabian Sea (Govil and Naidu, 2010). With enhanced tropical SST warming in East Indian Ocean, the latitudinal migration of the ITCZ potentially reduced, especially as the

seasonally-affected ITCZ generally locates over the warm ocean (Zhou et al., 2019) (**Lines 258-275**).

3) During the early deglaciation, ENSO characteristics change drastically in response to meltwater discharge and the resulting changes in the AMOC (Liu et al., 2014). Weakened AMOC lead to stronger ENSO activities, and a more El Niño-like state considering generally stronger El Niño than La Niña. The zonal Δ SST between West Pacific and East Pacific and model simulation suggest a more El Niño-like state in the early HS1(Liu et al., 2014, Koutavas and Joannides, 2012). Due to anomalous warming generated by El Niño under this state, the tropical troposphere becomes warmer, and subtropical troposphere is cooler, which enhances the meridional temperature gradient, and then results in shrinking of the Hadley circulation in both Hemispheres (**Lines 276-284**).

Generally speaking, we consider that the contracting precipitation belt was not affected by a single factor, but rather a result of multiple factors working together, including mainly related to the meltwater discharge, the resulting changes in the AMOC, the global temperature response and El Niño.

3. Minor comments. The manuscript would benefit from being read through by an English native speaker.

[Response]

Thank you for your meticulous evaluation of our manuscript.

We sincerely apologize for any errors that may have impeded your reading experience. We have taken the necessary steps to rectify these issues by thoroughly revising and rewriting the manuscript with the aid of the [Scientific English Rewriting](https://rewriting.ai/) website (<https://rewriting.ai/>). Our team has meticulously scrutinized each sentence for grammatical accuracy and scientific precision, ensuring that the manuscript meets the journal's standard.

Our deepest gratitude goes to reviewers for reviewers' careful work and thoughtful suggestions that have helped improve this paper substantially.

Reviewer #3 (Remarks to the Author):

This is the second time I review the manuscript by Yang et al., on the Intertropical Convergence Zone latitudinal changes and associated impacts on the tropical hydroclimate during Heinrich Stadial 1 (HS1), with emphasis on the Indian Ocean and the westernmost sector of the Pacific Ocean. The manuscript has greatly improved, notably the improved text (through revision of the English syntax) allows to follow the scientific content way better than in previous version. Accordingly, I would support acceptance in *Nature Communications*, pending revision related to the point detailed below.

In the previous assessment, my concerns centred on 3 main aspects of the study:

- (1) The lack of the Holocene section in the new palaeoceanographic time series presented for core I106, leaving the reader wondering about the actual amplitude of the discussed palaeoclimate signals with respect to the glacial-interglacial changes at the same site.
- (2) Role of the winds *versus* precipitation in the interpretation of the seawater $\delta^{18}\text{O}_{\text{SW}}$ changes discussed by the authors;
- (3) Relationship between Hadley cell circulation and global temperature.

Point 1 was somewhat addressed in the response letter but not in the manuscript. Figure R3 displays the *Globigerinoides ruber* $\delta^{18}\text{O}$ and the temperature gradient computed using *G. ruber* and (the thermocline species) *N. dutertrei* from core I106. Given that the data exist, my recommendation is to enrich with this information (*G. ruber* $\delta^{18}\text{O}$, Mg/Ca, and $\delta^{18}\text{O}_{\text{SW}}$) Figure 2 of the manuscript. This would allow highlighting the amplitude of the HS1 temperature and $\delta^{18}\text{O}_{\text{SW}}$ changes within the context of the full glacial-interglacial transition, so to provide a more complete analysis of the hydrographic changes in the study. In addition, it would help visualizing the $\delta^{18}\text{O}_{\text{SW}}$ signature of the Late Holocene relative to the early and middle Holocene. Note that Late Holocene $\delta^{18}\text{O}_{\text{SW}}$ data that are used to corroborate the relationship between $\delta^{18}\text{O}_{\text{SW}}$ and salinity given that "...estimates of $\delta^{18}\text{O}_{\text{SW}}$ values during the Late Holocene (0-2 ka) fall well within this linear $\delta^{18}\text{O}$ -salinity correlation". In conclusion, presentation of the full record for core I106 would make the analysis upon which the manuscript is centred more transparent and compelling.

Point 2 seems to be addressed, although some questions remain, I think that existing knowledge to agree with the authors' interpretation.

Point 3 has been satisfactorily addressed.

Response to REVIEWERS' COMMENTS

Reviewer #3 (Remarks to the Author):

This is the second time I review the manuscript by Yang et al., on the Intertropical Convergence Zone latitudinal changes and associated impacts on the tropical hydroclimate during Heinrich Stadial 1 (HS1), with emphasis on the Indian ocean and the westernmost sector of the Pacific Ocean. The manuscript has greatly improved, notably the improved text (through revision of the English syntax) allows to follow the scientific content way better than in previous version. Accordingly, I would support acceptance in *Nature Communications*, pending revision related to the point detailed below.

In the previous assessment, my concerns centred on 3 main aspects of the study:

(1) The lack of the Holocene section in the new palaeoceanographic time series presented for core I106, leaving the reader wondering about the actual amplitude of the discussed palaeoclimate signals with respect to the glacial-interglacial changes at the same site.

(2) Role of the winds *versus* precipitation in the interpretation of the seawater $\delta^{18}\text{O}_{\text{sw}}$ changes discussed by the authors;

(3) Relationship between Hadley cell circulation and global temperature.

Point 1 was somewhat addressed in the response letter but not in the manuscript. Figure R3 displays the *Globigerinoides ruber* $\delta^{18}\text{O}$ and the temperature gradient computed using *G. ruber* and (the thermocline species) *N. dutertrei* from core I106. Given that the data exist, my recommendation is to enrich with this information (*G. ruber* $\delta^{18}\text{O}$, Mg/Ca, and $\delta^{18}\text{O}_{\text{sw}}$) Figure 2 of the manuscript. This would allow highlighting the amplitude of the HS1 temperature and $\delta^{18}\text{O}_{\text{sw}}$ changes within the context of the full glacial-interglacial transition, so to provide a more complete analysis of the hydrographic changes in the study. In addition, it would help visualizing the $\delta^{18}\text{O}_{\text{sw}}$ signature of the Late Holocene relative to the early and middle Holocene. Note that Late Holocene $\delta^{18}\text{O}_{\text{sw}}$ data that are used to corroborate the relationship between $\delta^{18}\text{O}_{\text{sw}}$ and salinity given that "...estimates of $\delta^{18}\text{O}_{\text{sw}}$ values during the Late Holocene (0-2 ka)

fall well within this linear $\delta^{18}\text{O}$ -salinity correlation". In conclusion, presentation of the full record for core I106 would make the analysis upon which the manuscript is centred more transparent and compelling.

[Response]

We are grateful for this suggestion.

As suggested by the reviewer, we have added the Holocene data of *G. ruber* $\delta^{18}\text{O}$, Mg/Ca, Mg/Ca-SST and $\delta^{18}\text{O}_{\text{sw}}$ in **Figure 2** and Source Data file.

Point 2 seems to be addressed, although some questions remain, I think that existing knowledge to agree with the authors' interpretation.

[Response]

We appreciate the reviewer's positive evaluation of our work and agree with the comments regarding the limitations of our study.

Point 3 has been satisfactorily addressed.

[Response]

Our deepest gratitude goes to you for your careful work and thoughtful suggestions that have helped improve this paper substantially.